# REASONING SCAFFOLDING: DISTILLING THE FLOW OF THOUGHT FROM LLMS

**Xiangyu Wen**[1], **Junhua Huang**[2], **Zeju Li**[1], **Min Li**[3], **Jianyuan Zhong**[1], **Zhijian Xu**[1], **Mingxuan Yuan**[2], **Yongxiang Huang**[2], and **Qiang Xu**[*1,4]

[1]The Chinese University of Hong Kong
[2]Huawei Technologies Co., Ltd
[3]Southeast University
[4]Shenzhen Loop Area Institute
`{xywen22, zjli24, jyzhong, zjxu21, qxu}@cse.cuhk.edu.hk`
`{huangjunhua15, yuan.mingxuan, huang.yongxiang2}@huawei.com`

## ABSTRACT

The prevailing approach to distilling reasoning from Large Language Models (LLMs)—behavioral cloning from textual rationales—is fundamentally limited. It teaches Small Language Models (SLMs) to mimic surface-level patterns rather than the underlying algorithmic structure of thought, resulting in a critical lack of logical robustness. We argue that instead of cloning text, distillation should transfer this algorithmic structure directly. We introduce **Reasoning Scaffolding**, a framework that reframes reasoning as a structured generation process. Our method first abstracts the teacher's thought process into a sequence of discrete, interpretable semantic signals (e.g., `Contrast`, `Addition`) that act as a scaffold. The student model is then trained via a multi-task objective to both (1) *predict* the next semantic signal, anticipating the reasoning flow, and (2) *generate* the corresponding step, conditioned on that signal. This multi-task scheme acts as a powerful regularizer, compelling the student to internalize the computational patterns of coherent reasoning. On a suite of challenging reasoning benchmarks, our method significantly outperforms state-of-the-art distillation in both accuracy and logical consistency, providing a path towards creating smaller models that are genuine reasoners, not just fluent mimics[1].

## 1 INTRODUCTION

The prevailing approach to distilling reasoning from Large Language Models (LLMs)—behavioral cloning from Chain-of-Thought (CoT) rationales (Wei et al., 2022)—is fundamentally limited (Li et al., 2022; Ho et al., 2022; Shridhar et al., 2023). This method treats reasoning as a text imitation task, effectively forcing Small Language Models (SLMs) into a form of rote memorization (Gu et al., 2024; Anil et al., 2023). While this can teach stylistic fluency, it fails to transfer the underlying *algorithmic structure* of the teacher's thought process. Consequently, the resulting student models are often brittle, producing arguments that are logically inconsistent or nonsensical when faced with novel problems (Shridhar et al., 2023).

To move beyond this superficial mimicry, we argue for a paradigm shift: instead of teaching a model what to write, we must teach it how to think. Our key insight is that the teacher's reasoning process can be abstracted *from verbose text into its core structural blueprint*. This blueprint, composed of discrete, interpretable semantic signals like `Contrast` or `Elaboration`, governs the flow of a coherent argument. We introduce **Reasoning Scaffolding**, a new pedagogical framework that distills this structural blueprint, providing the student model with a scaffold to construct its own robust reasoning.

---

[*]Corresponding author.
[1]Our code is available at: https://github.com/xywen97/ReasoningScaffolding

Our framework implements this principle through a novel multi-task training regimen. We teach the SLM to reason step-by-step by having it simultaneously learn two crucial skills: (1) to *anticipate* the flow of a logical argument by predicting the next semantic signal, and (2) to *execute* a specific reasoning move by generating the corresponding text, conditioned on that signal. This dual objective, which uses the signal prediction task as a powerful regularizer for logical coherence, compels the student to internalize the computational patterns of reasoning rather than simply cloning text.

Our contributions are as follows:

- We introduce Reasoning Scaffolding, a new pedagogical framework that distills the structured, algorithmic flow of a teacher's reasoning, moving beyond surface-level text imitation.
- We propose a principled method for extracting and categorizing a 'reasoning scaffold' from textual rationales, creating a structured and interpretable training signal.
- We use a novel multi-task objective that forces the model to anticipate the logical function before generating the content.
- We demonstrate through extensive experiments on benchmarks like GSM8K (Cobbe et al., 2021) and StrategyQA (Geva et al., 2021) that our method yields SLMs that are significantly more accurate and logically robust than those trained with state-of-the-art distillation methods.
- We provide in-depth analysis showing closer alignment between the student and teacher's logical representations, providing strong evidence that our method produces genuine reasoners, not just fluent mimics.

In summary, this work establishes a theoretically grounded and empirically validated framework for transferring the core reasoning ability of LLMs, advancing the goal of creating powerful, efficient, and truly capable SLMs.

## 2 RELATED WORK

This section reviews the key research areas that underpin our work: (1) knowledge distillation in language models, (2) approaches for distilling reasoning capabilities, and (3) alternative paradigms for incorporating structure and logic into text generation.

### 2.1 KNOWLEDGE DISTILLATION FROM LLMS

Knowledge Distillation (KD) (Hinton et al., 2015; Fang et al., 2025; Xu et al., 2024) is a foundational technique for making the capabilities of large language models (LLMs) accessible and practical. The central goal is to transfer knowledge from a powerful, resource-intensive teacher model to a smaller, more efficient student model, thereby enabling wider deployment and application. Recent surveys (Xu et al., 2024; Zhu et al., 2024) show that KD in the LLM era now covers a broad spectrum of approaches, including model compression, skill transfer, and self-improvement for open-source models.

Recent advances have adapted core distillation algorithms to better suit the generative nature of LLMs. Traditional KD methods (Wang et al., 2025a) often use forward Kullback-Leibler (KL) divergence, which can cause the student to overestimate rare outputs from the teacher. To address this, Gu et al. (2024) propose using reverse KL divergence, which penalizes the student for generating outputs the teacher considers unlikely—proving more effective for generative tasks. Another major challenge is the architectural and vocabulary mismatch between teacher and student models. The Dual-Space Knowledge Distillation (DSKD) framework (Zhang et al., 2024) tackles this by aligning representations through a unified output space and cross-model attention, enabling distillation even when models use different tokenizers and vocabularies.

Beyond algorithmic improvements, recent work has focused on enhancing both the distillation process and the training data. For example, TinyLLM (Dai et al., 2024) introduces multi-teacher distillation, allowing students to learn from a diverse set of teacher LLMs and acquire a richer set of skills and reasoning patterns. Other approaches emphasize distilling not just final answers, but also

intermediate rationales or Chain-of-Thought steps, to impart deeper reasoning abilities (Ma et al., 2025). Research also shows that generating training data more efficiently—such as by incorporating student feedback into the teacher's output—can greatly reduce the need for large amounts of synthetic data. Additionally, curriculum-based fine-tuning schedules have been proposed to help SLMs internalize symbolic knowledge and perform complex reasoning without external tools at inference time (Liao et al., 2024).

However, despite these advancements, most existing KD methods focus on behavioral cloning from textual rationales. They prioritize transferring knowledge and stylistic fluency, while often overlooking the deeper reasoning capabilities and algorithmic paradigms that underpin robust logical thinking.

## 2.2 REASONING CAPABILITY DISTILLATION FROM LLMs

In addition to knowledge distillation, recent research seeks to transfer the advanced reasoning abilities of Large Language Models (LLMs)—especially those elicited by Chain-of-Thought (CoT) prompting (Wei et al., 2022; Wang et al., 2023b;a)—to smaller, more efficient models (SLMs) (Srivastava et al., 2025). The standard approach, reasoning distillation, involves fine-tuning SLMs on multi-step rationales generated by LLMs (Shridhar et al., 2023; Ho et al., 2022; Wang et al., 2025b). However, simply mimicking teacher rationales often leads SLMs to learn superficial patterns rather than the underlying logic, resulting in limited robustness.

To address this, recent works have focused on improving distillation data and methodology. For example, EDIT (Ho et al., 2022) uses pairs of similar but outcome-divergent reasoning traces to highlight critical inference steps, while other studies prune redundant steps from CoTs to promote concise, effective reasoning (Wang et al., 2025b; Li et al., 2025b). Mentor models (Lee et al., 2024), rationale decomposition (Xi et al., 2024), and modular architectures (Hinton et al., 2015) further enhance transfer by providing structured or higher-quality supervision.

However, most reasoning distillation methods overlook the underlying structural patterns present in extended reasoning traces—patterns that provide valuable signals for interpreting and guiding LLM reasoning. Li et al. (2025a) show that the structural coherence of reasoning chains, rather than the correctness of the plain content, is critical for enabling robust reasoning performance. This finding aligns with our insight that constructing a reasoning scaffolding can effectively guide an LLM's reasoning trace. However, while their work primarily emphasizes the importance of structural reasoning, we aim to advance this direction by developing a new distillation framework that first groups reasoning steps into abstract signals, thereby enabling data curation and explicit guidance of structural reasoning.

In the realm of interpretability, Concept Bottleneck LLMs (CB-LLMs) (Sun et al., 2025) introduce a framework where token decoding is made transparent through a concept bottleneck layer, allowing users to trace specific task concepts, such as 'Technology' and 'Business', while maintaining competitive accuracy. Yet, CB-LLMs primarily focus on token-level decoding in classification and plain text generation tasks. Building on this foundation, we extend the concept bottleneck approach by introducing discrete semantic reasoning signals into step-by-step, challenging reasoning tasks. This enables the transfer of algorithmic reasoning structure directly, rather than merely cloning textual rationales, addressing both interpretability and logical robustness in distillation.

## 3 METHOD

In this section, we detail the design and implementation of the Reasoning Scaffolding framework. Our approach comprises three key components: (1) **Logic Representation Distillation**, which prepares the training data by abstracting the teacher's reasoning process into structured semantic signals; (2) **Joint Training of Reasoning Proposer and Semantic Signal Predictor**, which enables the student model to anticipate and generate each step of reasoning; and (3) **Semantic Signal-Guided Reasoning at Inference**, which leverages the learned scaffold to guide the model's reasoning process during inference.

## 3.1 LOGIC REPRESENTATION DISTILLATION

We first query a large reasoning model (LRM, e.g., Deepseek-R1) using zero-shot prompting to obtain detailed reasoning traces (the prompt can be found in Appendix F.2). This yields a collection of long-thinking examples $S$, each comprising a step-by-step trace and a final summary. As shown in Figure 1, we can observe that certain keywords—such as 'wait', 'but', 'ok', and 'in addition'—naturally signal transitions in reasoning. For example, 'in addition' typically introduces supplementary information or elaboration. To systematically capture this phenomenon, we group these keywords into 7 semantic categories[2], such as 'Contrast and Concession', as semantic signals. We selected these 7 signals to ensure both **internal coherence**, where keywords within a group are semantically related (like 'in conclusion' and 'therefore'), and **external comprehensiveness**, where the categories collectively encompass the vast majority of transitions, as confirmed by manual review. A complete list of signals, along with a discussion of the signal curation strategy, is provided in Table 7 in Appendix 7 and further detailed in Appendix D.

| Step Content | Signals | All Semantic Signals |
|---|---|---|
| **... Additionally**, she bakes muffins for her friends every day with four eggs. That means she uses four eggs for muffins each day.\nAfter these uses, she sells the remainder at the farmers' market.\nI need to find out how many eggs are left to sell, and then multiply by the price per egg to find her daily earnings.\n\n | Addition and Elaboration | Addition and Elaboration |
| **... So**, step by step:\n1. Total eggs laid per day: 16\n2. Eggs eaten for breakfast: 3\n3. Eggs used for muffins: 4\n\n | Conclusion and Summary | Examples and Illustration |
| **... Then**, the eggs left for sale: total eggs minus eggs used = 16 - 7 = 9 eggs.\nNow, she sells these 9 eggs at $2 per egg.\n\n | Conclusion and Summary | Personal Opinion and Recall |
| **... But** let me calculate again.\nTotal eggs: 16\nMinus breakfast: 16 - 3 = 13 eggs left.\n\n | Contrast and Concession | Contrast and Concession |
| **... Perhaps** the question is about whether she sells all or part, but it says remainder.\nAnother thought: does she sell the eggs daily, meaning she might have eggs from previous days?\n\n | Personal Opinion and Recall | Reasoning and Analysis |
| **... Therefore**, the final answer must be 18.\n\n | Conclusion and Summary | Conclusion and Summary |
| **...** Summary of above reasoning content and generate responses | Response Generation | Response Generation |

Figure 1: Examples from GSM8K illustrating clear logical transition signals that guide reasoning direction.

Based on the above initial processing on the reasoning trace and semantic signals. We can build up our Reasoning Scaffolding dataset. We begin by segmenting the initial long-thinking trace into individual steps ($S_i = [A_1, \ldots, A_N]$), using double newline characters ('\n\n' or other characters that can separate the reasoning trace into individual steps) as delimiters[3]. As for generating the corresponding semantic signals for reasoning steps, we adopt a two-stage approach: First, we assign initial signal labels to each individual step based on keyword matches using Table 7. Second, we use a strong LLM (e.g., GPT-4.1) for semantic validation—verifying consistency between each step and its signal (the prompt example is in Appendix F.3). If a mismatch is detected, the LLM assigns the correct signal. For steps starting without signal keywords, the LLM directly provides the semantic label.

The hybrid keyword-LLM labeling strategy combines the efficiency of heuristics with the semantic understanding of a powerful LLM, resulting in a robust and highly efficient approach. By separating segmentation (structural) from labeling (using a hybrid of keywords and LLMs), we ensure that our scaffold accurately follows the teacher's flow, avoiding artificial fragmentation or omission of steps. The interplay and effectiveness of keyword matching versus LLM-based semantic labeling are further analyzed in Section 4.4.1 and Appendix E.4.

Using the step-by-step reasoning traces and the defined semantic signals, we generate two sets of training data: (1) pairs for the signal predictor, $\{Q + [A_1, \ldots, A_t], \text{Signal}\}$, for predicting next semantic signal, (2) pairs for the proposer model, $\{Q + [A_1, \ldots, A_t], \text{Signal} + A_{t+1}\}$, for generating next reasoning step. This process yields robust datasets for training both the signal predictor and the proposer model.

---

[2] 'Response Generation' marks the beginning of the Chain of Thought summary, while 'Conclusion and Summary' serves as the indicator for the 'Intermediate Summaries' within the reasoning trace.

[3] We consider only the signals that appear at the beginning of each step, preventing the creation of artificial or 'unneeded' steps.

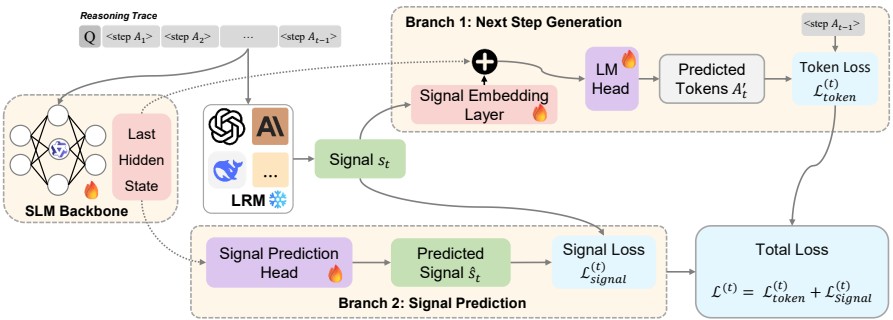

Figure 2: Reasoning Scaffolding framework. The dual-branch architecture trains an SLM to learn reasoning logic via two simultaneous tasks: signal-guided generation and signal prediction. The model is optimized to learn both the content and the structure of reasoning.

## 3.2 TRAINING SLMS AS STEP-BY-STEP PROPOSER AND LOGIC SIGNAL PREDICTOR

With the labeled data, we train the SLM to perform signal-guided, step-by-step reasoning. As shown in Figure 2, the modified model structure consists of a backbone and two branches.

The backbone is responsible for processing the input context, and producing the vectors of hidden states. In branch 1, we add an additional signal embedding layer (SEL) before the original language modeling (LM) head. The SEL layer is used to encode the pre-generated semantic signals $S_{i+1}$ into embeddings. The signal embeddings are fused with the backbone's last hidden state through simple addition and then passed to the LM head for next-token prediction. All tokens in a step share the same semantic signal, and training uses a modified next-token prediction loss:

$$\mathcal{L}^{(t)} = -\frac{1}{N_t} \sum_{i=1}^{N_t} \log P_\theta \left( A_{t,i} \mid A_{<t}, \ A_{t,<i}, \ s_t \right) \tag{1}$$

where $A_t$ represents tokens at step $t$ with length of $N_t$, $A_{<t}$ are previous steps, and $s_t$ is the associated semantic signal.

To further align the model's internal reasoning with the signal structure, we introduce a Signal Prediction Head as Branch 2. This branch compels the backbone to predict the current semantic signal, thereby increasing the model's sensitivity to signal cues and improving the semantic consistency between each reasoning step and its guiding signal. The signal prediction loss is defined as:

$$\mathcal{L}^{(t)}_{\text{signal}} = -\frac{1}{N_t} \sum_{i=1}^{N_t} \sum_{j=1}^{C} s_{t,i,j} \log P_\theta \left( \hat{s}_{t,i,j} \mid A_{<t} \right), \tag{2}$$

where $C$ is semantic signal number. The overall training objective is $\mathcal{L}^{(t)} = (1-\beta)\mathcal{L}^{(t)}_{\text{token}} + \beta\mathcal{L}^{(t)}_{\text{signal}}$.

In real-world applications, the signal predictor must generate next semantic signals in place of the ground-truth labels. The updated backbone provides a strong initialization for this predictor. For cold-start scenarios, we initialize a separate SLM with the trained backbone and then focus its training on signal prediction, further enhancing its ability to anticipate semantic signals during inference.

## 3.3 SEMANTIC SIGNAL-GUIDED REASONING

During inference, reasoning at each step is guided by predicted semantic signals. Given a question and the current reasoning trace, the signal predictor infers the most probable semantic signal for the next step. To ensure the correctness and reliability of these signals, we adopt an adaptive strategy, shown in Algorithm 1 in Appendix C. This approach begins by computing the confidence of the predicted signal:

$$\text{conf} = \exp \left( \frac{1}{L_t} \sum_{l=1}^{L_t} \log P_\phi(s_{t,l} \mid A_{<t}, s_{t,<l}) \right) \tag{3}$$

where $L_t$ is the length of signal $s_t$ at step $t$.

Predicted signals with confidence exceeding a threshold $\tau$ are used to guide the next reasoning step. If the confidence falls below $\tau$, the predicted signals are considered unreliable for further guidance. In such cases, we terminate the reasoning trace and prompt the model to generate a conclusion using the 'Response Generation' signal. The impact of the signal predictor's sensitivity to the hyperparameter $\tau$ is analyzed and illustrated in Table 5.

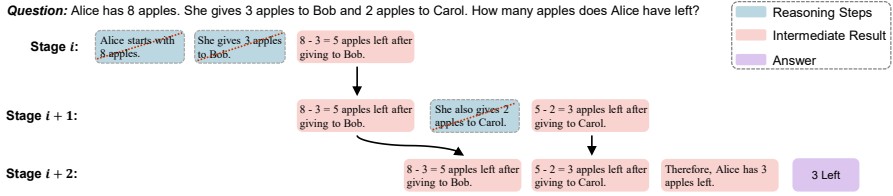

Figure 3: Token efficiency: By pruning reasoning steps before those labeled 'Conclusion and Summary', only intermediate results are retained.

Thanks to the interpretability of semantic signals, we can further optimize the reasoning trace. As illustrated in Figure 3, blocks in pink denote key intermediate results. We prune all other reasoning steps (blue blocks) within the same stage, retaining only these essential outputs. This process is repeated iteratively until the final answer (the purple block) is reached, significantly reducing token usage while preserving the critical information needed for downstream reasoning. Therefore, we adopt this pruning strategy as an optional, secondary optimization to address token efficiency.

## 4 EXPERIMENTS

### 4.1 EXPERIMENTAL SETTINGS

We evaluate our models on several distinct QA and mathematical benchmarks, each designed to target specific logical reasoning skills. **StrategyQA** (Geva et al., 2021) tests implicit multi-step reasoning, requiring the inference of necessary intermediate steps. **CommonsenseQA** (Talmor et al., 2019) assesses the application of common-sense knowledge in a multiple-choice format. **TruthfulQA** (Lin et al., 2021) measures a model's ability to avoid common misconceptions and imitative falsehoods. **GSM8K** (Cobbe et al., 2021) evaluates proficiency in grade-school mathematical problem solving, focusing on multi-step arithmetic and reasoning required to arrive at correct answers. Finally, **MATH** (Hendrycks et al., 2021; Lightman et al., 2023) assesses advanced mathematical reasoning and problem-solving skills across a broad range of topics, requiring models to generate detailed solutions to challenging competition-level math questions.

For evaluation, we use the Pass@1 metric to compare performance across different experimental settings throughout our paper. We experiment with a series of base models: Qwen-2.5-{0.5B, 7B, 14B}-instruct[4] (Yang et al., 2024).

### 4.2 MAIN RESULTS

We compare our method against several baselines implemented on different sizes of Qwen and Llama models. Specifically, the baselines include: the original pre-trained model, SFT models fine-tuned with Chain-of-Thought (CoT) and Long Thinking data, as well as Long-Thinking models distilled from Deepseek-R1.

The experimental results in Table 1 demonstrate the effectiveness of our method across a variety of question-answering and mathematical reasoning benchmarks. Compared to the original base models, our approach achieves a substantial improvement, with an average increase of approximately 14% in Pass@1 accuracy across all tasks. Furthermore, when compared to models fine-tuned with Chain-of-Thought (CoT) or Long-Thinking data, our method yields an average improvement of

---

[4]As of now, the 0.5B Deepseek distillation model has not been officially released.

Table 1: Main results (Pass@1)

| Methods | Models | StrategyQA | CommonsenseQA | TruthfulQA | GSM8K | MATH-500 |
|---|---|---|---|---|---|---|
| **Original** | Qwen2.5-0.5B-Ins | 0.543 | 0.475 | 0.268 | 0.379 | 0.335 |
| | Qwen2.5-7B-Ins | 0.726 | 0.785 | 0.706 | 0.875 | 0.738 |
| | Qwen2.5-14B-Ins | 0.755 | 0.785 | 0.750 | 0.921 | 0.764 |
| | Llama3.1-8B-Ins | 0.709 | 0.646 | 0.642 | 0.823 | 0.514 |
| **CoT SFT** | Qwen2.5-0.5B-Ins | 0.569 | 0.457 | 0.682 | 0.394 | 0.351 |
| | Qwen2.5-7B-Ins | 0.751 | 0.808 | 0.752 | 0.887 | 0.851 |
| | Qwen2.5-14B-Ins | 0.760 | 0.810 | 0.831 | 0.928 | 0.882 |
| **Long Thinking SFT**[a] | Qwen2.5-0.5B-Ins | 0.571 | 0.463 | 0.670 | 0.412 | 0.388 |
| | Qwen2.5-7B-Ins | 0.759 | 0.817 | 0.771 | 0.862 | 0.879 |
| | Qwen2.5-14B-Ins | 0.768 | 0.845 | 0.812 | 0.931 | 0.901 |
| **Long Thinking Distill**[b] | Qwen2.5-7B-Ins | 0.680 | 0.670 | 0.544 | 0.867 | 0.883 |
| | Qwen2.5-14B-Ins | 0.811 | 0.805 | 0.763 | 0.936 | 0.904 |
| **GPT-oss-120B** | Few-Shot | 0.783 | 0.825 | 0.862 | 0.768 | 0.872 |
| **Deepseek-R1** | Few-Shot | **0.863** | **0.895** | **0.874** | **0.961** | **0.965** |
| **Ours (Teacher=GPT-oss)** | Qwen2.5-7B-Ins | 0.748 | 0.822 | 0.868 | - | 0.850 |
| **Ours (Teacher=DS-R1)** | Llama3.1-8B-Ins | 0.783 | 0.841 | 0.843 | 0.864 | 0.844 |
| | Qwen2.5-0.5B-Ins | 0.659 | 0.601 | 0.861 | 0.488 | 0.417 |
| | Qwen2.5-7B-Ins | 0.832 | 0.866 | 0.879 | 0.899 | 0.922 |
| | Qwen2.5-14B-Ins | 0.858 | 0.887 | 0.917 | 0.942 | 0.928 |

[a] Official distilled model released by Deepseek.
[b] Our SFT model trained on long-thinking data distilled and generated by Deepseek-R1.

about 8%. We also observe a consistent trend of performance improvement as the model size increases. Our method is effective across all model scales, from the smallest 0.5B model to the largest 14B model, highlighting its strong generalization ability to different model capacities and diverse benchmarks. Further discussion on the experimental results can be found in Appendix E.1 and Appendix E.3.

Notably, our approach brings significant gains to smaller models. For instance, on the TruthfulQA dataset, the 0.5B model's Pass@1 accuracy increases dramatically from around 27% (original) and 68% (CoT SFT) to over 86% with our method. This demonstrates the potential of our approach to substantially enhance reasoning and factual correctness even in resource-constrained settings.

### 4.3 COMPARISON OF REASONING TRACES

To explicitly demonstrate the advantages of our Reasoning Scaffolding method over the baselines, we compare the reasoning traces generated by different models, including the original model, the CoT-SFT model, and our model enhanced with Reasoning Scaffolding.

Figure 4 presents detailed reasoning traces produced by different models. These cases illustrate both why and how our approach improves the model's reasoning abilities.

In Column 1, the reasoning trace generated by the original model contains clear mistakes. For example, the model incorrectly treats 'cool off' as an object and selects it as a possible answer. Although it correctly identifies that 'ice cream' is sweet, it fails to establish the connection between 'ice cream' and 'enjoy something,' resulting in an incorrect final answer. In Column 2, the CoT-SFT model also fails to predict the correct answer and continues to consider 'cool off' as an option. While it demonstrates some improvement by recognizing that 'ice cream' is both sweet and cool, it still does not link eating 'ice cream' with the concept of 'enjoying something.'

In contrast, our Reasoning Scaffolding method (Column 3) enables the model to correctly identify that 'cool off' is not an object, and successfully explains why 'ice cream' is the correct answer. This is achieved by learning to adhere to the logical flow in Column 4 (e.g., Reasoning and Analysis → Conclusion and Summary → Reasoning and Analysis), our model is guided to produce a coherent and correct argument, which prevents it from making the logical errors that the baseline models fall into. These results highlight how our approach strengthens the model's ability to perform logical reasoning and connect relevant concepts, leading to more accurate and interpretable outcomes.

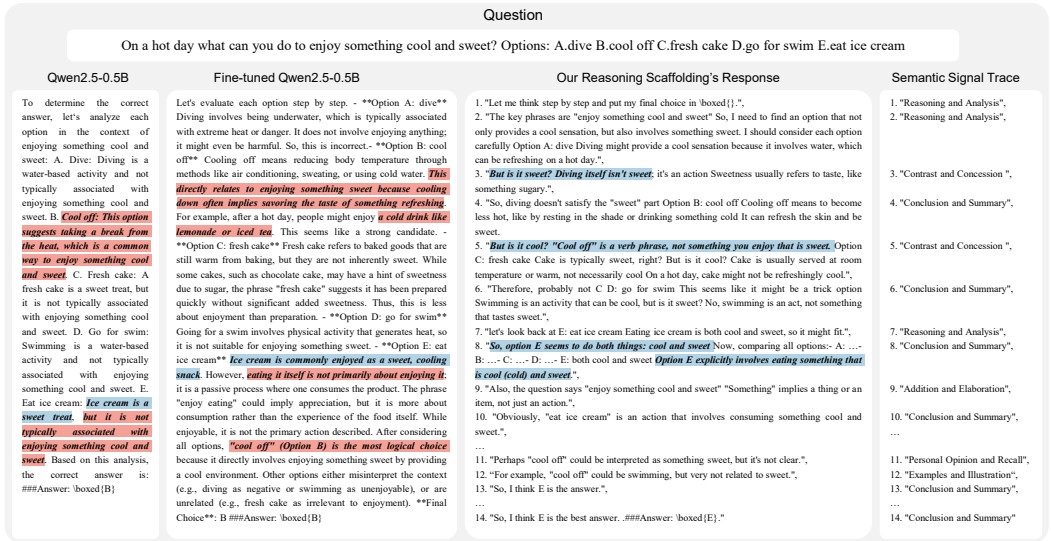

Figure 4: Case studies analyzing how semantic signal-guided reasoning enhances the model's reasoning capability.

## 4.4 ABLATION STUDY

In this section, we conduct an ablation study to investigate the contributions of the key components of our Reasoning Scaffolding method, specifically focusing on the hyper-parameter sensitivity analysis on the signal prediction and the impact of signal quality on reasoning performance. For clarity, we present some of the experimental results solely on the 14B model; additional results for other model scales can be found in Appendix E.6 and Appendix E.7.

### 4.4.1 SIGNAL PREDICTOR

To enhance the applicability of our method, we train a signal predictor using data derived from golden semantic signals. The training data consists of pairs in the format <current context, next signal>, constructed as described in Section 3.1. Detailed examples of this training data can be found in Appendix G. We employ supervised fine-tuning for the signal predictor, utilizing the same base model as the proposer, as outlined in Section 3.2.

As described in Section 3.1, we annotate reasoning steps using a two-stage process that combines keyword matching with LLM-based verification. Empirically, we find that approximately 74% of all reasoning steps begin with one of our predefined keywords. For these keyword-initiated steps, the labeling accuracy—defined as the proportion of steps for which the keyword-based semantic signal matches the LLM's semantic labeling—is about 87%. The remaining 26% of reasoning steps, which do not start with any predefined keyword, are labeled directly using the LLM oracle (GPT-4.1). Further details on the labeling procedure are provided in Appendix E.4.

Table 2: Signal prediction accuracy.

| Model Size | StrategyQA | CommonsenseQA | TruthfulQA | GSM8K | MATH-500 | Averaged |
|---|---|---|---|---|---|---|
| Qwen 0.5B | 0.748 | 0.739 | 0.732 | 0.729 | 0.737 | 0.737 |
| + Adaptive | **0.783** | **0.777** | **0.791** | **0.778** | **0.784** | **0.783** |
| Qwen 7B | 0.791 | 0.788 | 0.802 | 0.801 | 0.796 | 0.796 |
| + Adaptive | **0.839** | **0.841** | **0.837** | **0.835** | **0.837** | **0.838** |
| Qwen 14B | 0.841 | 0.843 | 0.841 | 0.829 | 0.836 | 0.838 |
| + Adaptive | **0.857** | **0.849** | **0.855** | **0.858** | **0.849** | **0.854** |

As shown in Table 2, the signal predictor achieves a next-signal prediction accuracy exceeding 75%, and this accuracy increases to over 83% when the base model scale is expanded to 14B. Detailed

accuracy rates for each individual signal are reported in Table 14 in the Appendix. Furthermore, by incorporating the adaptive signal prediction strategy introduced in Section 3.3, the Pass@1 accuracy for next-signal prediction rises to above 85%, indicating that reliable and accurate semantic signals can be provided to guide the decoding of proposer models.

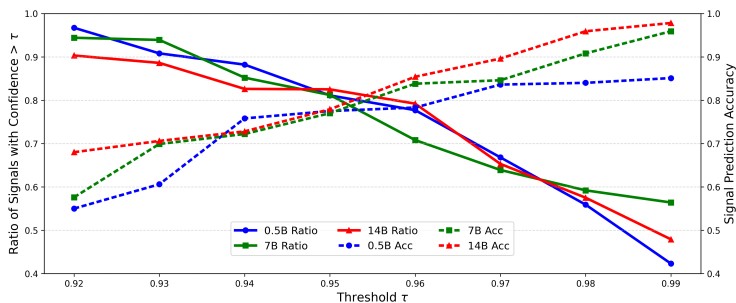

Figure 5: The effects of threshold $\tau$ on the ratio of predicted signals with log probability $> \tau$ and the signal prediction accuracy.

Since the adaptive signal prediction strategy is controlled by the threshold parameter $\tau$, we further investigate its impact on prediction accuracy. As illustrated in Figure 5, there is a noticeable intersection between the curves representing signal prediction accuracy and the ratio of signals with confidence $> \tau$ for $\tau$ values between 0.95 and 0.96. To strike a balance between retaining a higher proportion of predicted signals and maintaining high prediction accuracy, we set the global value of $\tau$ to 0.96 throughout our experiments.

### 4.4.2 SIGNAL QUALITY AND REASONING PERFORMANCE

Table 3 illustrates the impact of signal quality on the reasoning performance of the proposer model. Specifically, we conduct three sets of experiments, where the proposer is guided by golden signals, signals generated by the signal predictor, and randomly generated signals. The results clearly show that the quality of signals significantly influences the model's reasoning capabilities.

Table 3: Accuracy analysis under different signal strategies.

| Benchmarks | Original | w/ Golden Signals | w/ Signal Predictor | w/ Random Signals | Summaries Only |
|---|---|---|---|---|---|
| StrategyQA | 0.755 | **0.858** | 0.843 | 0.776 | 0.855 |
| CommonsenseQA | 0.785 | **0.887** | 0.869 | 0.827 | 0.869 |
| TruthfulQA | 0.750 | **0.917** | 0.885 | 0.828 | 0.897 |
| GSM8K | 0.921 | **0.942** | 0.933 | 0.929 | 0.941 |
| MATH-500 | 0.764 | **0.928** | 0.918 | 0.894 | 0.916 |

When using golden signals, the proposer achieves the highest accuracy across all benchmarks. There is a slight decrease in performance when switching to signals predicted by the signal predictor; however, the accuracy remains substantially higher than the original model. This indicates that the signal predictor is able to reproduce most of the high-quality semantic signal traces.

The strong performance of the 'Summaries Only' strategy highlights a key insight: the intermediate conclusions ('Conclusion and Summary' steps) contain the most vital information for the reasoning process. While the full spectrum of signals provides the best performance, this finding suggests that focusing on conclusion states is a primary driver of the gains. This is also consistent with the observation that humans can continue reasoning based on previous intermediate results to arrive at the final answer.

Interestingly, using random signals still provides a benefit over standard fine-tuning. This suggests that the mere act of structuring the generation into discrete steps—even with a nonsensical scaffold—serves as a useful inductive bias, preventing the model from defaulting to monolithic text imitation. However, using the correct, golden signals provides a significant additional gain, demonstrating that learning the correct reasoning flow is crucial for optimal performance. We also present a failure case where the proposer is guided by shuffled semantic signals, as shown in Appendix E.9.

### 4.4.3 TOKEN CONSUMPTION ANALYSIS

Table 4 reports the token consumption across different methods and benchmarks. As expected, token usage increases from CoT-based methods to long-thinking approaches. Our Reasoning Scaffolding method, when guided by all signals, produces reasoning traces with token lengths comparable to those of long-thinking traces distilled from large reasoning models, yet achieves significantly higher reasoning accuracy (see Table 1). We provide a more detailed discussion of both training and inference computational overhead in Appendix E.7.

Table 4: Token consumption analysis under different strategies.

| Methods | StrategyQA | CommonsenseQA | TruthfulQA | GSM8K | MATH-500 |
|---|---|---|---|---|---|
| Original | 224 | 217 | 282 | 304 | 604 |
| CoT SFT | 438 | 392 | 509 | 343 | 715 |
| Thinking Distill | 597 | 613 | 705 | 771 | 2,616 |
| Thinking SFT | 1,878 | 1,959 | 1,877 | 1,996 | 5,921 |
| **Ours + All Signals** | **1,524** | **1,638** | **1,550** | **1,659** | **4,755** |
| **- Remove Reasoning Steps** | **746** | **823** | **773** | **845** | **2,628** |

We acknowledge that our Reasoning Scaffolding method, in its current form, generates longer reasoning traces compared to standard CoT. This reflects a trade-off where our approach prioritizes maximal reasoning fidelity and logical coherence over token efficiency. The token reduction strategy in Section 3.3 helps mitigate this, the length of reasoning traces can be significantly shortened, while still maintaining high accuracy (refer to the last column of Table 3). Therefore, we use the pruning strategy as an optional, secondary optimization to address token efficiency. future work could explore methods for pruning the scaffold to create more compact yet equally effective reasoning paths.

## 5 LIMITATIONS AND FUTURE WORK

Despite the promising results and novel perspective introduced in this work, several limitations merit discussion for a balanced and transparent account.

First, our approach primarily extracts high-level discourse markers (e.g., Contrast, Addition, Conclusion) rather than formal algorithmic or logical operations. Thus, our 'reasoning scaffolding' acts as a tractable and interpretable proxy for reasoning structure, but does not directly encode programmatic or symbolic computation. Future work could address this by incorporating more fine-grained logical or algorithmic signals to bridge the gap between discourse-level and formal reasoning.

Second, our annotation methodology achieves scalability by using heuristic techniques (e.g., segmentation by double newlines, keyword matching) for most cases, with LLM-based oracle validation (GPT-4.1) applied only as needed. While effective, this still introduces some dependency on the oracle LLMs, potentially affecting scalability and reproducibility. Future research could focus on developing more robust, self-supervised signal extraction methods. We would also like to explore extending our framework to additional scenarios, such as creative writing and planning, to demonstrate the generalization capability of our method.

Overall, these limitations do not diminish the core contributions of this work but instead highlight promising directions for future research. Addressing these challenges will further improve the scalability, robustness, and practical utility of reasoning scaffolds in language models.

## 6 CONCLUSION

We proposed Reasoning Scaffolding, a novel distillation framework that empowers Small Language Models to internalize the structured reasoning patterns of Large Language Models. Unlike conventional rationale-based approaches, our method distills semantic signals rather than surface-level text, effectively addressing key limitations in existing techniques. This leads to student models that are not only more accurate, but also exhibit greater logical consistency and interpretability. Extensive experiments across multiple benchmarks demonstrate the effectiveness of our approach, highlighting its potential as a promising direction for efficient and faithful knowledge transfer in language model distillation.

ACKNOWLEDGMENTS

This work was supported in part by the CUHK Research Matching Scheme under Grant No. 7106937 and 8601130.

ETHICS STATEMENT

All experiments and analyses presented in this paper strictly adhere to the ICLR Code of Ethics. The models and datasets utilized in this work are publicly available, fully open-source, and do not involve proprietary or restricted data. No human subjects, private or sensitive data, or human labor were involved in the development, training, or evaluation of our methods. Our research does not raise concerns related to privacy, security, discrimination, bias, fairness, or legal compliance.

REPRODUCIBILITY STATEMENT

To ensure the reproducibility of our work, we have provided comprehensive details of the model architectures, dataset descriptions, and training procedures in the Experiments section and the Appendix. All datasets and models used in our study are open-source, and we have released our code alongside this paper to further facilitate reproducibility.

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

## A   THE USE OF LARGE LANGUAGE MODELS

We use Large Language Models (LLMs), including ChatGPT and Gemini, solely for polishing the writing of this paper.

## B   IMPLEMENTATION DETAILS

Table 5 summarizes the models and datasets used in our study, including three Qwen2.5-Instruct variants evaluated on five benchmarks with varying train and test set sizes.

Table 5: Details of models and datasets.

| Items | Values | | | | |
|---|---|---|---|---|---|
| **Models** | Qwen2.5-0.5B-Ins | Qwen2.5-7B-Ins | Qwen2.5-14B-Ins | | |
| **Benchmarks** | StrategyQA | CommonsenseQA | TruthfulQA | GSM8K | MATH |
| **Train Set Size** | 1,602 | 9,741 | 657 | 7,473 | 7,500 |
| **Test Set Size** | 687 | 1,140 | 159 | 1,319 | 500 |

We selected our suite of benchmarks (e.g., GSM8K, StrategyQA, TruthfulQA) specifically because they test fundamentally different types of reasoning:

- GSM8K/MATH require formal, algorithmic, and arithmetic reasoning.
- StrategyQA/CommonsenseQA require implicit, multi-step commonsense reasoning.
- TruthfulQA tests factual correctness and the avoidance of imitative falsehoods.

As shown in Table 5, the dataset sizes range from hundreds to thousands, demonstrating the strong robustness and generalization ability of our method. In addition, to improve the quality of the datasets used for training our model, we propose a two-stage data annotation method (see Section 3.1). This method combines a keyword-matching approach with LLM validation, which further enhances signal correctness and overall data quality (see Table 8 for details).

Table 6: Training details.

| Parameter | Value | Description |
|---|---|---|
| **Model Name** | Qwen/Qwen2.5-{0.5B, 7B, 14B}-Instruct | Base model |
| **Learning Rate** | 1e-4/5e-5 | Initial learning rate |
| **Batch Size** | 1 | Per-device batch size |
| **Num Train Epochs** | 3∼5 | Number of training epochs |
| **Gradient Accum. Steps** | 8 | Gradient accumulation steps |
| **PEFT / LoRA** | True (r=16, $\alpha$=16) | Adapter fine-tuning (LoRA) |
| **LR Scheduler Type** | Cosine | Learning Rate Scheduler Type: (Cosine or Linear) |
| **Completion only Loss** | True | Compute loss on the completion segment only |
| **Warmup Ratio** | 0.05 | Warmup Ratio |
| **Loss Weighting** | 0.1 - 0.9 | Balance two losses |

Table 7 presents all the keywords found in the long-thinking trace along with their corresponding semantic signal names.

Table 7: All keywords in the long-think trace and their corresponding semantic signal names

| Semantic Signals | Keywords | | | | | | |
|---|---|---|---|---|---|---|---|
| **Contrast and Concession** | but
still | however
although | on the other hand
whereas | otherwise | nevertheless | nonetheless | in contrast |
| **Addition and Elaboration** | also | moreover | additionally | furthermore | in addition | | |
| **Examples and Illustration** | for example | for instance | | | | | |
| **Personal Opinion and Recall** | i think
i recall | i believe
i remember | i guess | in my opinion | maybe | it seems | perhaps |
| **Reasoning and Analysis** | first
i'll/need | actually
let's see | in fact
wait | let me
ok | anyway
well | by the way
now | of course |
| **Conclusion and Summary** | so
similarly | then
unless | after all
as a result | obviously
therefore | clearly
thus | indeed
to conclude | meanwhile
in conclusion |

In addition to the aforementioned semantic signals, we introduce another signal, 'Response Generation', to indicate the initiation of summary generation for the reasoning trace. From a content

perspective, this signal corresponds to the chain-of-thought segment following the '</think>' markers. The 'Response Generation' signal is also utilized for early stopping and for directly generating the response summary, as discussed in Section 3.3.

## C PSEUDO-CODE OF ADAPTIVE DECODING ALGORITHM

---

**Algorithm 1** Semantic Signal-Guided Adaptive Reasoning

---

**Require:** Proposer model $P_\theta$, Signal Predictor $P_\phi$, confidence threshold $\tau$, initial Question $Q$
**Ensure:** Final Reasoning Trace $T = [A_1, A_2, ..., A_t]$

1: $T \leftarrow [Q]$
2: **while** true **do**
3:                                                                  $\triangleright$ Predict the next signal
4:         $\hat{s}_t, \text{conf}_t \leftarrow P_\phi(T_{<t})$                                   $\triangleright$ using Equation 3
5:                                                $\triangleright$ Adapt based on confidence
6:         **if** $\text{conf}_t < \tau$ **then**
7:             $s_t \leftarrow$ "Response Generation"
8:         **else**
9:             $s_t \leftarrow \hat{s}_t$
10:       **end if**
11:                                                    $\triangleright$ Generate the next step
12:       $A_t \leftarrow P_\theta(T_{<t}, s_t)$
13:       $T \leftarrow T + [A_t]$
14:                                   $\triangleright$ Stop if Chain-of-Thought conclusion is generated
15:       **if** $s_t =$ "Response Generation" **then**
16:            **break**
17:       **end if**
18: **end while**
19: **return** $T$

---

## D DISCUSSION ON REASONING SCAFFOLDING METHOD

In this section, we discuss the generalization ability and validity of our Reasoning Scaffolding method, focusing on the selection of semantic signals and the method's extensibility.

The seven categories are not chosen arbitrarily, nor are they specific to any particular model or task. They derive from a bottom-up, iterative manual review of the teacher's long-thinking traces, as described in Section 3.1. We begin by identifying common keywords, and then group them into semantically coherent categories (e.g., 'but', 'however', 'in contrast' all become 'Contrast and Concession'). Our goal is to create a set that: 1) collectively "encompasses the vast majority of transitions"; 2) contains keywords within a group that are semantically related; and 3) remains small and distinct enough for an SLM to learn effectively. These semantic signals are universally present in coherent thought, regardless of the domain, because they correspond to fundamental discourse relations. A legal argument or code explanation follows a similar structural flow. Therefore, we argue these are not "GSM8K-specific patterns," but rather the core features of any logical argument.

Our experimental results strongly support the claim that these signals generalize across different areas. We evaluate the exact same 7-signal scaffold on TruthfulQA and StrategyQA. As shown in Table 1, our method outperforms baselines significantly on these non-math tasks (e.g., +11% over CoT SFT on TruthfulQA for the 7B model). This demonstrates that the taxonomy successfully captures the reasoning flow in diverse domains (fact-checking, commonsense) without modification of the semantic signals.

We also clarify that the "Reasoning Scaffolding" contribution is the distillation framework itself (the multi-task objective and signal-guided inference), not the static list of 7 signals. While we find these 7 to be robustly effective across our diverse benchmarks, the framework is inherently flexible. For highly specialized domains (e.g., multi-modal reasoning or competitive programming), the framework allows for the definition of domain-specific signal sets (e.g., adding a 'Visual Interpretation' or 'Debugging' signal) using the same extraction pipeline described in Section 3.1.

Overall, our method reframes the distillation objective: first, the model predicts the logical function of the next step (e.g., Contrast, Conclusion); then it generates text to realize that function. Rather than simple text imitation, this approach distills control flow, teaching the student model the algorithmic blueprint of reasoning. The key insight is that our multi-task objective serves as a strong regularizer for logical coherence. By requiring the model to predict a semantic signal before generating text, we compel it to internalize the computational patterns of reasoning—transforming it from a fluent mimic into a structured reasoner.

# E EXPERIMENTAL RESULTS

## E.1 MAIN RESULTS ANALYSIS

Table 8: Main results (Pass@1) - duplicated from Table 1

| Methods | Models | StrategyQA | CommonsenseQA | TruthfulQA | GSM8K | MATH-500 |
|---|---|---|---|---|---|---|
| | Qwen2.5-0.5B-Ins | 0.543 | 0.475 | 0.268 | 0.379 | 0.335 |
| Original | Qwen2.5-7B-Ins | 0.726 | 0.785 | 0.706 | 0.875 | 0.738 |
| | Qwen2.5-14B-Ins | 0.755 | 0.785 | 0.750 | 0.921 | 0.764 |
| | Qwen2.5-0.5B-Ins | 0.569 | 0.457 | 0.682 | 0.394 | 0.351 |
| CoT SFT | Qwen2.5-7B-Ins | 0.751 | 0.808 | 0.752 | 0.887 | 0.851 |
| | Qwen2.5-14B-Ins | 0.760 | 0.810 | **0.831** | 0.928 | 0.882 |
| | Qwen2.5-0.5B-Ins | 0.571 | 0.463 | 0.670 | 0.412 | 0.388 |
| Long Thinking SFT[a] | Qwen2.5-7B-Ins | 0.759 | 0.817 | 0.771 | 0.862 | 0.879 |
| | Qwen2.5-14B-Ins | 0.768 | **0.845** | 0.812 | 0.931 | 0.901 |
| Long Thinking Distill[b] | Qwen2.5-7B-Ins | 0.680 | 0.670 | 0.544 | 0.867 | 0.883 |
| | Qwen2.5-14B-Ins | **0.811** | 0.805 | 0.763 | **0.936** | **0.904** |
| | Qwen2.5-0.5B-Ins | 0.659 | 0.601 | 0.861 | 0.488 | 0.417 |
| **Ours** | Qwen2.5-7B-Ins | 0.832 | 0.866 | 0.879 | 0.899 | 0.922 |
| | Qwen2.5-14B-Ins | **0.858** | **0.887** | **0.917** | **0.942** | **0.928** |

[a] Official distilled model released by Deepseek.
[b] Our SFT model trained on long-thinking data distilled and generated by Deepseek-R1.

Based on our evaluation of the Deepseek Distill Qwen models, in most cases, we observed that the Distill model consistently under-performs compared to the SFT model trained on long thinking data—and even falls short of the raw Qwen2.5 base model. For example, on StrategyQA, the Distill model (0.680) lags behind both the SFT (0.759) and base model (0.726); similarly, on CommonsenseQA, it achieves 0.670 versus 0.817 (SFT) and 0.785 (base). This performance gap is particularly pronounced on QA benchmarks. We hypothesize that this drop in performance may be attributed to the distillation process being heavily optimized for a different task distribution, with a primary focus on areas such as mathematics and code. While the Distill model achieves competitive results in mathematical reasoning—such as 0.867 on GSM8K and 0.883 on MATH-500, similar to or even slightly higher than the SFT and base models—this appears to come at the expense of general and commonsense reasoning. We believe these findings further underscore the motivation for our reasoning scaffolding method, as they suggest that standard distillation methodologies can be brittle and may fail to uniformly transfer reasoning capabilities across diverse tasks.

## E.2 ABLATION ON LOSS WEIGHTING

We utilize two distinct loss functions during training: a next token prediction loss for text generation and a cross-entropy loss for semantic signal prediction. Observations throughout our experiments indicate that both losses maintain similar magnitudes, allowing us to adopt a simple 1:1 weighting scheme for implementation. Across extensive trials on models of varying sizes (0.5B, 7B, and 14B), we did not encounter any training instability. Our approach consistently delivered notable improvements, as evidenced in Table 1, where our method surpasses all baseline models on five challenging benchmarks. The signal prediction loss acts as an effective regularizer, increasing the model's responsiveness to external signals; this straightforward combination has reliably produced the performance gains reported.

To further evaluate the impact of loss weighting, we conducted an ablation study using the TruthfulQA dataset and the Qwen2.5-7B model, varying the loss weighting parameter ($L^t = (1 - \beta) * L^t_{token} + \beta * L^t_{signal}$) The results show that adjusting $\beta$ between 0.3, 0.5, and 0.7 yields nearly identical accuracy scores, demonstrating stable performance across different weightings. Increasing

Table 9: Ablation on Loss Weighting Parameter

| | β | | |
|---|---|---|---|
| | **0.3** | **0.5** | **0.7** |
| **Accuracy** | 0.868 | 0.874 | 0.874 |
| **Token Loss Convergence Steps** | 277/500 | 305/500 | 326/500 |
| **Signal Loss Convergence Steps** | 261/500 | 258/500 | 229/500 |

β leads to slightly faster signal loss convergence and slower token loss convergence, but these shifts do not affect overall accuracy. This indicates a minor trade-off in convergence behavior without introducing instability or compromising results. Consequently, our default 1:1 loss weighting is both robust and efficient, as further tuning does not yield meaningful improvements. These findings reinforce the effectiveness and reliability of our approach under varying loss weight configurations.

### E.3 BENCHMARKING TEACHER MODELS

Here, we benchmark the two teacher models of 'Deepseek-R1' and 'GPT-oss-120B', the experimental results are shown in Table 11.

Table 10: Distribution of semantic signals frequency of GPT-oss and Deepseek-R1

| Semantic Signals | GPT-oss | Deepseek-R1 |
|---|---|---|
| Contrast and Concession | 0.142 | 0.286 |
| Addition and Elaboration | 0.065 | 0.043 |
| Examples and Illustration | 0.001 | 0.146 |
| Personal Opinion and Recall | 0.022 | 0.036 |
| Reasoning and Analysis | 0.559 | 0.294 |
| Conclusion and Summary | 0.103 | 0.171 |
| Response Generation | 0.108 | 0.024 |

We first calculate the semantic signal frequency within the reasoning traces generated by GPT-oss and Deepseek-R1. As shown in the table below, the two models exhibit different biases in their reasoning styles. GPT-oss tends to favor forward reasoning steps, although reflection and intermediate summary steps are still clearly present. Interestingly, traces generated by GPT-oss contain very few Example and Illustration steps. In contrast, Deepseek-R1 demonstrates a more balanced distribution among reflection, addition, personal opinion, and forward reasoning steps.

Table 11: Benchmarking Teacher Models

| Methods | Models | StrategyQA | CommonsenseQA | TruthfulQA | GSM8K | MATH-500 |
|---|---|---|---|---|---|---|
| | Directly Answer | 0.840 | 0.855 | 0.811 | 0.955 | 0.939 |
| Deepseek-R1 | Zero-Shot | 0.855 | 0.870 | 0.818 | 0.957 | 0.958 |
| | Few-Shot | **0.863** | **0.895** | **0.874** | **0.961** | **0.965** |
| | Directly Answer | 0.762 | 0.805 | 0.767 | 0.755 | 0.908 |
| GPT-oss-120B | Zero-Shot | **0.823** | 0.808 | 0.780 | 0.761 | **0.920** |
| | Few-Shot | 0.783 | **0.825** | **0.862** | **0.768** | 0.872 |

### E.4 SIGNAL LABELING CORRECTNESS

Here we show more details on the data generation accuracy for signal prediction, mainly focusing on the combination of the keyword-based and LLM-based method. We investigate the agreement between keyword-based categorization and LLM-based labeling for splitting traces into categories. The results are presented in Table 12. Specifically, we report two key metrics: Keyword Covering Rate and Keyword Labeling Correctness.

- Keyword Covering Rate measures the proportion of traces that contain at least one relevant keyword, indicating the effectiveness of our keyword selection for each dataset.
- Keyword Labeling Correctness evaluates the agreement between keyword-based labels and those assigned by the LLM, reflecting how well keyword labeling matches the LLM's understanding.

Table 12: Keyword covering rate and labeling correctness.

|  | StrategyQA | CommonsenseQA | TruthfulQA | GSM8K | MATH |
|---|---|---|---|---|---|
| **Keyword Covering Rate** | 0.771 | 0.715 | 0.687 | 0.743 | 0.798 |
| **Keyword Labeling Correctness** | 0.850 | 0.825 | 0.922 | 0.874 | 0.903 |

As shown in the table, the covering rates range from 0.687 to 0.798, demonstrating that most of the reasoning steps are start with the explicit keywords defined in Table 7 in the Appendix. This also means that our keyword-matching method is valid for step labeling. The labeling correctness is consistently high across all datasets, with values from 0.825 to 0.922, indicating strong agreement between keyword-based and LLM-based validation.

These results suggest that our keyword-based approach for trace categorization is both comprehensive and reliable, closely aligning with LLM judgments and supporting the validity of our analysis.

We clarify that the around 8-17% disagreement rate refers to the initial keyword heuristic relative to the LLM. It does not represent the error rate of the final training dataset. As described in Section 3.1 , our data generation pipeline is corrective. When the keyword heuristic disagrees with the LLM, we adopt the LLM's label as the ground truth for training. The student model is trained on the cleaner, LLM-validated signals, not the raw, noisy keyword predictions. Therefore, the 'error accumulation' concern does not apply to the training phase. To address the concern about the reliability of the two methods given the disagreement, we conduct a manual human verification on a random sample of 50 reasoning steps where the Keyword and LLM disagreed (the 'error' cases). Below is some cases:

Table 13: Cases for checking labeling correctness.

| Reasoning Steps | Keyword-matching | LLM-validation | Human-validation |
|---|---|---|---|
| Well, is Madrid considered a big city? Yes, it's the capital of Spain. | Reasoning and Analysis | Addition and Elaboration | Addition and Elaboration |
| Perhaps the key is to check if cousins are mentioned. | Personal Opinion and Recall | Reasoning and Analysis | Reasoning and Analysis |
| But just to be thorough, is there any chance Marie Antoinette said something similar? I don't think so. Biographers and historians agree that it's a myth. | Contrast and Concession | Personal Opinion and Recall | Personal Opinion and Recall |
| Meanwhile, for C and D, it's implied. | Conclusion and Summary | Addition and Elaboration | Addition and Elaboration |
| But compared to 100%, it might be, but I think in this context, it's meant to be incorrect. | Contrast and Concession | Personal Opinion and Recall | Contrast and Concession |

The experimental results in Table 13 show that the human annotation matches the LLM's label in 96% of cases, confirming that the 'disagreement' is primarily due to the limited context of the keyword heuristic, and that the LLM acts as a highly reliable oracle. Therefore, we believe that combining the two labeling methods—keyword matching and the LLM oracle—can significantly improve labeling accuracy. Keyword matching offers rapid initial labeling at virtually no cost, while the LLM acts as an oracle for final validation and as an auxiliary tool for the remaining steps. By integrating these approaches, we can generate high-quality oracle training data, leading to a more reliable and cleaner training dataset.

### E.5 DETAILED SIGNAL PREDICTION ACCURACY

We also conducted additional experiments to evaluate the prediction accuracy for each individual signal in detail. Table 14 presents the detailed prediction accuracy for each signal using different base models for training the signal predictor.

Initially, alongside the current seven signal categories, we include an additional category called 'Discourse Markers and Fillers', which comprised words such as 'well', 'actually', and 'wait'. However, our signal prediction experiments show that this category is often misclassified as 'Reasoning and Analysis', 'Personal Opinion and Recall', and other existing categories, as shown in the confusion

Table 14: Signal prediction accuracy of each signal.

| Signal Predictor | Reasoning and Analysis | Addition and Elaboration | Examples and Illustration | Personal Opinion and Recall | Contrast and Concession | Conclusion and Summary | Response Generation | Average |
|---|---|---|---|---|---|---|---|---|
| Qwen2.5-0.5B-Instruct | 0.745 | 0.661 | 0.851 | 0.693 | 0.748 | 0.822 | 0.943 | 0.783 |
| Qwen2.5-7B-Instruct | 0.815 | 0.689 | 0.896 | **0.767** | 0.802 | 0.848 | 0.979 | 0.838 |
| Qwen2.5-14B-Instruct | **0.835** | **0.721** | **0.912** | 0.757 | **0.816** | **0.875** | **0.998** | **0.854** |

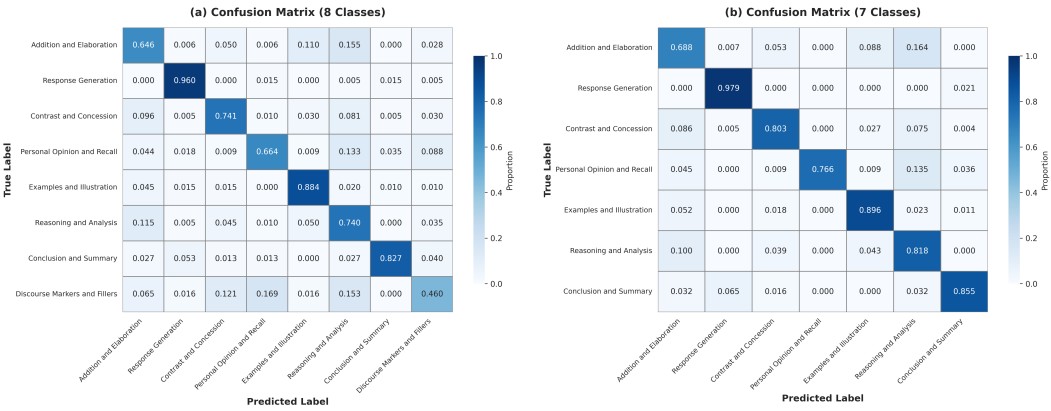

Figure 6: Confusion matrix comparison before and after adjusting semantic signals.

matrix of Figure 6. In this case, to improve clarity and guidance, we have since reassigned these keywords to the relevant groups within the seven established signal categories.

### E.6 SIGNAL QUALITY AND REASONING PERFORMANCE (ON 0.5B AND 7B MODEL)

To further validate the impact of signal quality on reasoning performance, we replicate the main experiments on both a smaller 0.5B parameter model and a larger 7B parameter model. Tables 15 and 16 report the accuracy results under five signal strategies across all benchmarks.

Table 15: Accuracy analysis under different signal strategies (0.5B Model).

| Benchmarks | Original | w/ Golden Signals | w/ Signal Predictor | w/ Random Signals | Summaries Only |
|---|---|---|---|---|---|
| StrategyQA | 0.543 | **0.659** | 0.632 | 0.583 | 0.64 |
| CommonsenseQA | 0.475 | **0.601** | 0.587 | 0.459 | **0.601** |
| TruthfulQA | 0.267 | **0.861** | 0.834 | 0.676 | 0.855 |
| GSM8K | 0.379 | **0.488** | 0.459 | 0.39 | 0.478 |
| MATH-500 | 0.335 | **0.417** | 0.393 | 0.362 | 0.401 |

Results show that, similar to the main model, the use of golden signals and predicted signals significantly improves accuracy over the original baseline. Notably, even randomly generated signals and summaries-only strategies yield consistent gains, particularly on TruthfulQA and StrategyQA, suggesting the robustness of the Reasoning Scaffolding approach even in smaller models.

The trends closely mirror those observed in the main experiments, with golden signals yielding the highest accuracy across benchmarks. The 7B model demonstrates overall stronger performance, and signal-guided reasoning consistently boosts accuracy, highlighting the scalability and effectiveness of semantic signal guidance across model sizes.

### E.7 TRAINING AND INFERENCE OVERHEAD ANALYSIS (ON 0.5B AND 7B MODELS)

To assess the efficiency of Reasoning Scaffolding across different model scales, we report token consumption results for both the 0.5B and 7B parameter models under various reasoning strategies. Tables 17 and 18 present the average token usage per benchmark for each method.

Table 16: Accuracy analysis under different signal strategies (7B Model).

| Benchmarks | Original | w/ Golden Signals | w/ Signal Predictor | w/ Random Signals | Summaries Only |
|---|---|---|---|---|---|
| StrategyQA | 0.726 | **0.832** | 0.797 | 0.757 | 0.799 |
| CommonsenseQA | 0.785 | **0.866** | 0.841 | 0.823 | 0.858 |
| TruthfulQA | 0.706 | 0.879 | 0.859 | 0.752 | **0.881** |
| GSM8K | 0.875 | **0.899** | 0.873 | 0.829 | 0.882 |
| MATH-500 | 0.738 | **0.922** | 0.883 | 0.866 | 0.908 |

Table 17: Token consumption analysis under different strategies (0.5B Model).

| Methods | StrategyQA | CommonsenseQA | TruthfulQA | GSM8K | MATH-500 |
|---|---|---|---|---|---|
| Original | 180 | 192 | 259 | 271 | 523 |
| CoT SFT | 499 | 344 | 449 | 321 | 679 |
| Thinking SFT | 1797 | 1927 | 1847 | 1971 | 5652 |
| **Ours + All Signals** | **1447** | **1501** | **1432** | **1696** | **4673** |
|   - Remove Reasoning Steps | **727** | **749** | **616** | **793** | **2235** |

Table 18: Token consumption analysis under different strategies (7B Model).

| Methods | StrategyQA | CommonsenseQA | TruthfulQA | GSM8K | MATH-500 |
|---|---|---|---|---|---|
| Original | 216 | 213 | 276 | 320 | 553 |
| CoT SFT | 464 | 364 | 491 | 355 | 704 |
| Thinking Distill | 530 | 572 | 651 | 686 | 2422 |
| Thinking SFT | 1840 | 2072 | 1856 | 2013 | 5841 |
| **Ours + All Signals** | **1579** | **1587** | **1467** | **1790** | **5103** |
|   - Remove Reasoning Steps | **758** | **819** | **792** | **767** | **2422** |

The results highlight that Reasoning Scaffolding (Ours + All Signals) produces longer reasoning traces than standard CoT approaches, but remains more efficient than Thinking SFT. Further token savings are achieved by removing reasoning steps after intermediate conclusions, demonstrating flexibility in controlling output length while maintaining reasoning performance. Compared to the long-thinking method, our Reasoning Scaffolding method can achieve higher performance with a much shorter reasoning trace.

The trends are consistent with those observed in smaller models: Reasoning Scaffolding delivers detailed reasoning with moderate token growth, and applying the step-removal strategy significantly reduces token usage. This confirms the scalability and efficiency of the approach across model sizes.

In addition, we also expand our discussion on the training computational overhead of our method and its comparison with the CoT and Long-Thinking distillation approaches.

Table 19: Training computational overhead analysis.

| | Original Training Dataset Size | Signal Prediction | | Token Prediction | | Prefilling |
|---|---|---|---|---|---|---|
| | | Average Signal Length /tokens | Dataset Size /samples | Average Length /tokens | Average Dataset Size /samples | Average Length /tokens |
| CoT Distillation | 657 - 9,741 | N/A | N/A | 515 | 657 - 9,741 | 123 |
| Long Thinking Distill | 657 - 9,741 | N/A | N/A | 1,935 | 657 - 9,741 | 135 |
| **Ours** | **657 - 9,741** | **3.8** | **4,195 - 19,317** | **90** | **4,195 - 19,317** | **827** |

Table 19 provides a detailed comparison of the computational overhead among CoT distillation, Long Thinking Distill, and our proposed dual-branch training. Because our method operates in a step-by-step manner, it produces a greater number of training samples (4,195–19,317 for signal and token prediction) compared to the other approaches. However, each sample is much shorter: the average token count for signal prediction is only 3.8, and for next-step token prediction, it is 90—both significantly less than those in CoT distillation and Long Thinking Distill. This trade-off enables our method to maintain data efficiency and reduce token-level computation, resulting in overall lower computational overhead during training and inference.

### E.8 TOKEN LENGTH VS. PERFORMANCE:

To conclusively determine if the performance gap is merely due to increased token length, we conduct an additional experiment where we enforce the "Thinking Distill" baseline to generate extended reasoning traces (matching the length of our method) via prompt constraints. The results (see Table 20) explicitly refute the hypothesis that "longer is better" for this baseline:

Table 20: Reasoning scaffolding enhances the model's performance by promoting structured reasoning, rather than simply increasing token length.

| Items | StrategyQA | CommonsenseQA | TruthfulQA | GSM8K |
|---|---|---|---|---|
| **Accuracy of Original Version** | 0.811 | 0.805 | 0.763 | 0.936 |
| **Accuracy of Enforced Long Thinking** | 0.783 ($\downarrow$) | 0.805 (-) | 0.724 ($\downarrow$) | 0.915 ($\downarrow$) |
| **Accuracy of Ours** | 0.858 | 0.887 | 0.917 | 0.942 |
| **Thinking Length of Original Version** | 597 | 613 | 705 | 771 |
| **Thinking Length of Enforced Long Thinking** | 1,789 | 1,617 | 1,876 | 1,865 |
| **Thinking Length of Ours** | 1,524 | 1,638 | 1,550 | 1,659 |

As shown in Table 20, When forced to reason for longer durations, the accuracy of "Thinking Distill" actually stagnated or even dropped. This demonstrates that token length acts as a liability for the baseline; without structural guidance, the model "gets lost" in the extended context, leading to hallucinations or loss of focus. In contrast, our Reasoning Scaffolding achieves high accuracy because of the structured semantic signals, which allow the model to utilize the longer context effectively without losing coherence. Thus, the performance gap is driven by the scaffold, not the token count.

### E.9 REASONING TRACES GUIDED BY RANDOM SIGNALS

Below, we present a comparison between reasoning traces guided by the original semantic signal order and those guided by a random order.

The key difference between the two reasoning traces is that the correct answer's reasoning is grounded in accurate legal facts about the U.S. criminal justice system, specifically that only the prosecutor (not the victim) can drop criminal charges once filed, and each step checks assumptions against legal reality. In contrast, the first reasoning trace relies on personal belief and incorrect assumptions—mistakenly thinking the victim has the authority to drop charges—without verifying this against actual legal procedure. The first reasoning fails because it builds its logic on a fundamental factual error about who controls criminal prosecution, never fact-checking this critical point.

---

**Question**

In the U.S. can the victim of a domestic violence case drop the charges? Options: A. Yes, the victim can drop the charges B. Yes, if the victim is the accuser, they are free to drop the charges C. No, the victim cannot drop the charges once a prosecutor is involved

---

**Original Semantic Signals**

Reasoning and Analysis, Reasoning and Analysis, Contrast and Concession, Personal Opinion and Recall, Contrast and Concession, Contrast and Concession, Personal Opinion and Recall, Contrast and Concession, Reasoning and Analysis, Examples and Illustration, Conclusion and Summary, Contrast and Concession, Personal Opinion and Recall, Contrast and Concession, Reasoning and Analysis, Personal Opinion and Recall, Conclusion and Summary, Conclusion and Summary, Conclusion and Summary, Conclusion and Summary, Contrast and Concession

---

**Random Semantic Signals**

Personal Opinion and Recall, Examples and Illustration, Conclusion and Summary, Contrast and Concession, Contrast and Concession, Personal Opinion and Recall, Conclusion and Summary, Reasoning and Analysis, Personal Opinion and Recall, Contrast and Concession, Conclusion and Summary, Contrast and Concession, Conclusion and Summary, Contrast and Concession, Conclusion and Summary, Conclusion and Summary, Reasoning and Analysis, Contrast and Concession, Reasoning and Analysis, Contrast and Concession, Reasoning and Analysis

## Reasoning Trace Guided by Random Signals

1. Personal Opinion and Recall: I think I remember that in the US, the victims of domestic violence have the right to drop charges.

2. Examples and Illustration: For example, in many jurisdictions, the prosecution cannot overcharge or force the case to trial if the victim doesn't want it.

3. Conclusion and Summary: So, option A seems correct.

4. Contrast and Concession: But let's look at option B: Yes, if the victims is the accused, they are free to drop the charges That seems redundant because if the same person is both the victims and the accused, it might be confusing, but essentially, they can still drop charges.

5. Contrast and Concession: But option C says: "No, the victims cannot drop the charges once a prosecuted is involved" This might be tricky

6. Personal Opinion and Recall: I recall that when a case is prosecuted, the prosecution can apply for a warrant or move the case along, but the final decision to drop charges is often with the victims, especially in domestic violence cases where there might be a lot of personal details.

7. Conclusion and Summary: I think A is correct.

8. Reasoning and Analysis: Wait, let me double-check.

9. Personal Opinion and Recall: I think in some states, there might be specific laws, but generally, the principle is that the victims have a say in the case Option B says if the victims is the accused, which might mean the same person is both, but it's poorly worded It should probably be if the victim is the accused, but even then, it's not clear.

10. Contrast and Concession: But in context, it's probably meant to be the same as A.

11. Conclusion and Summary: So, I think A is the best answer.

12. Contrast and Concession: But let's see why C might be tempting. Sometimes, if the charge is serious, like if a warrant is out, the prosecution might not allow it, but for domestic violence, it's common to have the option to drop.

13. Conclusion and Summary: Therefore, I think A is correct.

14. Contrast and Concession: But just to be thorough, I remember that in some cases, like if the crime is perjury or something, but for domestic violence, it's fine.

15. Conclusion and Summary: So, my final answer should be A.

16. Conclusion and Summary: So, choose A.

17. Reasoning and Analysis: OK, I think that's it.

18. Contrast and Concession: But let's see the options again.

19. Reasoning and Analysis: Actually, A Yes, the victims can drop the charges B Yes, if the victims is the accused, they are free to drop the charges – this might imply that in some cases, the accused is also the victims, but it's awkward phrasing C No, the victims cannot drop the charges once a prosecuted is involved – this suggests that once involved, they can't drop, which is false.

20. Contrast and Concession: But in reality, they can.

21. Reasoning and Analysis: OK, A is correct.

## Reasoning Trace Guided by Godel Signals

1. Reasoning and Analysis: First, the question is: "In the U.S. can the victim of a domestic violence case drop the charges?" The options are:A. Yes, the victim can drop the charges B. Yes, if the victim is the accuser, they are free to drop the charge C. No, the victim cannot drop the charges once a prosecutor is involved.

2. Reasoning and Analysis: I need to recall how criminal charges work in the U.S. especially for domestic violence. In criminal cases, it's not the victim who files charges; it's the state or the government. The victim might report the crime or be a witness, but the prosecution is handled by a prosecutor, like a district attorney. For domestic violence cases, victims often want to drop charges because of fear, reconciliation, or other reasons.

3. Contrast and Concession: But can they actually drop the charges themselves?

4. Personal Opinion and Recall: I remember that once charges are filed, the victim doesn't have the authority to drop them. The prosecutor decides whether to proceed with the case or not. This is because domestic violence is considered a crime against the state, not just the individual victim. Option C says: "No, the victim cannot drop the charges once a prosecutor is involved." That sounds familiar.

5. Contrast and Concession: But let me check options A and B. Option A says "Yes, the victim can drop the charges," which seems too broad and probably incorrect. Option B says "Yes, if the victim is the accuser, they are free to drop the charges."

6. Contrast and Concession: But in criminal cases, the accuser is typically the state, not the victim. The victim might be the complainant, but not the one prosecuting. In some jurisdictions, victims can file for restraining orders or civil actions, but criminal charges are separate.

7. Personal Opinion and Recall: I think there's a common misconception that victims can drop charges, but in reality, they cannot. Prosecutors can consider the victim's wishes, but they are not bound by them. Prosecutors might proceed even if the victim wants to drop charges, especially in domestic violence cases, to prevent coercion or further harm.

8. Contrast and Concession: But is that universal? I think it varies, but generally, victims cannot unilaterally drop charges.

9. Reasoning and Analysis: Let me think about the legal process. When a crime is reported, the police investigate, and if there's enough evidence, the prosecutor files charges. The victim becomes a witness. The victim doesn't have the power to dismiss the case; only the prosecutor or the court can do that.

10. Examples and Illustration: For example, if a victim refuses to testify, the prosecution might have a harder time, but they can still proceed with other evidence or subpoena the victim.

11. Conclusion and Summary: So, option C seems correct: "No, the victim cannot drop the charges once a prosecutor is involved."

12. Contrast and Concession: But what about before charges are filed? If the victim reports domestic violence but later decides not to pursue it, can they prevent charges from being filed? In some cases, if no charges have been filed yet, the victim might not cooperate with the investigation, which could lead to no charges. But once charges are filed, it's out of their hands. The question says "drop the charges," implying that charges have already been filed. Option B mentions "if the victim is the accuser." In criminal law, the accuser is the state, not the individual. So that might be misleading. In some contexts, like private prosecutions, but I think in the U.S. criminal cases are almost always public, with state prosecutors.

13. Personal Opinion and Recall: I think C is the correct answer.

14. Contrast and Concession: But let me double-check. Is there any scenario where a victim can drop charges? For instance, in some misdemeanor cases or with consent, but for domestic violence, it's often treated more seriously.

15. Reasoning and Analysis: Actually, upon second thought, in some jurisdictions, for certain offenses, victims might have more say, but generally for felonies and serious crimes, no. Domestic violence can be a misdemeanor or felony, but still, the principle holds.

16. Personal Opinion and Recall: I remember a key point: the victim is not a party to the criminal case; the case is The State v. Defendant, not Victim v. Defendant. So the victim doesn't control the prosecution.

17. Conclusion and Summary: Therefore, the victim cannot drop charges.

18. Conclusion and Summary: So, options A and B are incorrect. Option B says "if the victim is the accuser," but they aren't the accuser in the legal sense; the prosecutor is.

19. Conclusion and Summary: So, I think C is correct. Final answer should be C. The question is multiple choice, and I need to box it.

20. Conclusion and Summary: So, \boxedC

21. Contrast and Concession: But let me make sure. Is there any trick here? The question says "in the U.S." and laws can vary by state, but generally, this is true across states. Yes, I think it's standard.

# F    PROMPT DESIGN

## F.1    PROMPTS FOR BENCHMARKING

Below is a series of prompts used to benchmark the base models for direct answering, zero-shot prompting, and few-shot prompting.

---

### Prompts for Directly Answering Prompting Strategy for Benchmarking

Answer the following single-choice question directly and putting your choice in ###Answer:\boxed{} without any explanations.
This is the question:
{question}

Your choice:

---

### Prompts for Zero-shot Prompting Strategy for Benchmarking

Answer the following single-choice question step-by-step and putting your choice in ###Answer:\boxed{}.
This is the question:
{question}

Your solution:

---

### Prompts for Few-shot Prompting Strategy for Benchmarking

Answer the following single-choice question step-by-step and putting your choice in ###Answer:\boxed{}.
This is the question:
What is the most common cause of the Bermuda Triangle disappearances? Options: \nA. The Bermuda Triangle disappearances are caused by a mysterious alien force.\nB. The Bermuda Triangle disappearances are the result of a curse placed on the region.\nC. There is no single cause, but factors like magnetic anomalies, human error, and severe weather have contributed to the disappearances.\nD. The Bermuda Triangle disappearances are caused by a large underwater vortex.

Your solution:
Let me answer this question about the Bermuda Triangle disappearances step by step:\n\n1) First, I need to consider what the Bermuda Triangle is. It's a region in the western part of the North Atlantic Ocean where ships and aircraft have allegedly disappeared under mysterious circumstances.\n\n2) Let me evaluate each option:\n\n - Option A suggests alien forces are responsible. This is a popular theme in science fiction but lacks scientific evidence.\n\n - Option B suggests a curse is responsible. This is a supernatural explanation without scientific basis.\n\n - Option C suggests multiple natural factors including weather, human error, and magnetic anomalies. This aligns with scientific investigations.\n\n - Option D suggests an underwater vortex. While ocean currents exist, there's no evidence of a specific vortex causing disappearances.\n\n3) Scientific investigations of incidents in the Bermuda Triangle have found:\n - Many disappearances can be explained by severe weather like hurricanes and waterspouts\n - Human error in navigation\n - Equipment failure\n - Some compass variations due to natural magnetic properties of the region\n - Many incidents occurred during conditions of poor visibility\n\n4) Additionally, the U.S. Coast Guard and other scientific organizations have stated that the number of disappearances in the Bermuda Triangle is not statistically higher than other busy shipping areas when accounting for the high volume of traffic.\n\n5) The most logical explanation supported by evidence is that multiple natural factors, rather than a single supernatural cause, explain the various incidents.\n\nTherefore, the answer is: C.\n\n###Answer: \boxed{C}

Answer the following single-choice question step-by-step and putting your choice in ###Answer:\boxed{}.
This is the question:
Does the popular saying 'You can't teach an old dog new tricks' have any basis in science? Options: \nA. The saying is based on the fact that older dogs' brains shrink, making learning impossible.\nB. The saying is true, older dogs can only learn new tricks with intense training.\nC. The saying is a misconception, as research shows that older dogs can still learn new tricks, though it may take more time and patience.\nD. The saying is true, older dogs are incapable of learning new tricks.

Your solution:
Let me approach this question about whether "You can't teach an old dog new tricks" has any basis in science.\n\nStep 1: Analyze what science tells us about aging and learning in dogs.\nResearch in canine cognition has shown that while aging does affect cognitive function in dogs (as in humans), older dogs maintain the ability to learn. Their brains don't simply "shrink" and become incapable of forming new neural connections. Cognitive decline can occur, but it doesn't make learning "impossible."\n\nStep 2: Consider how age affects learning in dogs.\nStudies have found that older dogs may:\n- Learn more slowly than younger dogs\n- Require more repetitions\n- Need more patience during training\n- Benefit from shorter training sessions\n- Experience some decline in certain cognitive functions\n\nHowever, these changes don't prevent learning altogether - they just change the optimal approach to teaching.\n\nStep 3: Evaluate the scientific evidence for each option.\nA: Incorrect - While some brain atrophy can occur with age, it doesn't make learning "impossible."\nB: Incorrect - Older dogs don't require "intense" training; they require appropriate, patient training.\nC: Aligns with scientific evidence - Research does show older dogs can learn, though potentially with modified approaches.\nD: Incorrect - This contradicts scientific evidence on canine cognition.\n\nStep 4: Draw a conclusion.\nThe saying "You can't teach an old dog new tricks" is indeed a misconception. Science shows that while aging affects learning processes, older dogs maintain the ability to learn throughout their lives. The key difference is that it may take more time, patience, and appropriate teaching methods.\n\nTherefore, the answer is: C\n\n###Answer: \boxed{C}

Answer the following single-choice question step-by-step and putting your choice in ###Answer:\boxed{}.
This is the question:

---

> {question}
>
> Your solution:

## F.2 Prompts for Generating Long-Thinking Data

We directly prompt the DeepSeek-R1 model to obtain the long-thinking trace as the initial dataset.

> ### Prompts for Generating Initial Data
>
> Answer the following math question step by step and put your final choice in \n ###Answer: \\boxed{}
>
> This is the question: {question}
>
> Your solution:

## F.3 Prompts for Labeling Reasoning Traces by Semantic Meaning

To enhance the correctness and reliability of semantic signals as labels for reasoning steps, we also employ LLMs as judges to verify whether the logic signals are assigned appropriately to each reasoning step.

> ### Prompts for Semantic Signal Checking
>
> You are a judge who evaluates the semantic logic transition between a new sentence/paragraph (the "text") and its preceding context. Select the transition type strictly from the following list:
>
> 1. Contrast and Concession: The new sentence or paragraph presents a contrast or concession in relation to the previous context.
> 2. Addition and Elaboration: The new sentence or paragraph supplements or expands upon the previous context.
> 3. Examples and Illustration: The new sentence or paragraph provides examples or explanations related to the previous context.
> 4. Personal Opinion and Recall: The new sentence or paragraph primarily expresses a personal viewpoint or belief.
> 5. Reasoning and Analysis: The new sentence or paragraph serves as a starting point for reasoning or doing analysis based on the current information.
> 6. Conclusion and Summary: The new sentence or paragraph concludes from the previous context and present an intermediate result. outcome.
>
> ## Instructions: Given the previous context and the new sentence, analyze the semantic logic transition from the context to the text. Your answer should contain only the transition type, formatted as: \boxed{logic transition name}
>
> ## Context:
> {context}
>
> ## Text to be judged:
> {text}
>
> ## Your Response:

## F.4 Prompts for Signal Prediction

Here is the prompt for training a signal predictor, mainly including the semantic signals and the task description for signal prediction.

> ### Prompts for Signal Predictor
>
> Given the following context from a reasoning or problem-solving process, predict the most likely semantic signal (from a predefined set of signal groups) to which the next reasoning keyword or phrase will belong.
>
> The possible semantic signals are:
>
> 1. Contrast and Concession
> 2. Addition and Elaboration
> 3. Examples and Illustration
> 4. Personal Opinion and Recall
> 5. Reasoning and Analysis
> 6. Conclusion and Summary
> 7. Response Generation

Please answer by putting the semantic signal name in \\boxed{semantic signal name}.

This is the reasoning context:
{context}

The semantic signal is:

## G  TRAINING EXAMPLES

Below, we provide examples used for training and testing the Proposer and the Signal Predictor. Specifically, for training the Proposer, the input is the question, and we update the model's parameters by computing both the token loss—between the gold reasoning step and the predicted step—and the signal loss—between the gold signal and the predicted signal—using the two branches of the modified model architecture. The details in the following examples include only the long-thinking trace within <think> and </think>, without the final answer generation guided by the Response Generation signal, which serves as a standard Chain-of-Thought block.

### Example-1

#### Question

What should you do if you have a nosebleed? Options: A. You should lie down B. You should lean forward and breathe through your mouth C. You should tilt your head back

| Step-by-Step Reasoning Trace | Semantic Signals |
|---|---|
| 1. First, the question is: "What should you do if you have a nosebleed?" with three options: A. You should lie down B. You lean forward and breathe through your mouth C. You should tilt your head back. | 1. Reasoning and Analysis, |
| 2. I need to recall the correct first aid for nosebleeds. I've heard different advice, but I think the standard recommendation has changed over time. | 2. Reasoning and Analysis, |
| 3. Actually, I remember that tilting your head back used to be a common suggestion, but it's not recommended anymore. Why? Because if you tilt your head back, the blood can run down your throat, which might cause choking or swallowing blood, and that can lead to nausea or vomiting. | 3. Reasoning and Analysis, |
| 4. So, option C: "You should tilt your head back" – that might be incorrect. | 4. Conclusion and Summary, |
| 5. Now, option A: "You should lie down." Is lying down good for a nosebleed? I think lying down could also cause blood to flow down the throat, similar to tilting the head back. It might increase pressure in the veins of the nose, making bleeding worse. Option B: This sounds familiar. Leaning forward helps the blood drain out through the nose instead of down the throat. Breathing through the mouth prevents you from breathing in blood or causing more irritation. | 5. Reasoning and Analysis, |
| 6. I think the American Red Cross or Mayo Clinic recommends sitting upright and leaning forward slightly. Pinching the soft part of the nose is also key, but that's not mentioned in the options. The question is about what to do, and option B includes leaning forward and breathing through the mouth. | 6. Personal Opinion and Recall, |
| 7. Let me confirm: Yes, standard advice is to sit down, lean forward, pinch the nostrils together for about 10 minutes, and breathe through your mouth. | 7. Reasoning and Analysis, |
| 8. So, option B seems correct. | 8. Conclusion and Summary, |
| 9. But let's double-check option A and C. Option A: Lie down – no, that's bad because it can cause blood to go into the stomach or lungs. Option C: Tilt head back – same issue, blood goes down the throat. Option B: Lean forward – good, blood comes out, and you can spit it out if needed. | 9. Contrast and Concession, |
| 10. Also, breathing through the mouth makes sense to avoid nasal passages. | 10. Addition and Elaboration, |
| 11. I think B is the correct answer. | 11. Personal Opinion and Recall, |
| 12. But is there any scenario where you might lie down or tilt back? I don't think so for typical first aid. Medical professionals might have other methods, but for general public, leaning forward is key. The question says "you should do," implying general advice. | 12. Contrast and Concession, |
| 13. So, I'll go with B | 13. Conclusion and Summary, |
| 14. So, final answer should be B. | 14. Conclusion and Summary, |

## Example-2

### Question

What does it always mean when a cat purrs? Options: A. A cat's purr always means that the cat is hungry B. A cat's purr always means that the cat is stressed C. A cat's purr can signify many emotions, including both happiness and anxiety D. A cat's purr always means that the cat is happy

### Step-by-Step Reasoning Trace

1. First, the question is: "What does it always mean when a cat purrs?" The options are: A. A cat's purr always means that the cat is hungry B. A cat's purr always means that the cat is stressed C. A cat's purr can signify many emotions, including both happiness and anxiety D. A cat's purr always means that the cat is happy

2. I need to choose the correct one and box my answer at the end. Key point: The word "always" is crucial here. It says "always mean," so I need to find which statement is universally true for all purring instances.

3. I recall that cats purr for various reasons. From what I know, purring isn't just about one emotion. Cats purr when they're content, like when they're being petted or resting comfortably. But they also purr when they're in pain, stressed, or even when giving birth. I think purring can be a self-soothing mechanism.

4. For example, a cat might purr when it's happy and relaxed, not necessarily hungry. Or it might purr when injured, which isn't happiness or hunger.

5. I remember reading that kittens purr when nursing, which might indicate contentment, but adult cats purr in various situations.

6. Also, some studies suggest that purring can promote healing or reduce stress, so it's not just about positive emotions.

7. So, option C seems correct because it acknowledges that purring can mean different things, not just one.

8. But the question says "always mean," and C says "can signify," which might not directly answer "always." Let me read the options again. Option C: "A cat's purr can signify many emotions..." – this implies that it doesn't always mean one thing; it can vary.

9. Whereas A, B, D claim that it always means one specific thing.

10. But based on my knowledge, purring doesn't always mean hunger, stress, or happiness. It depends on the context.

11. Therefore, A, B, and D are incorrect because they are absolute statements that aren't true. C is the only one that correctly states that purring can signify multiple emotions. The question is phrased as "what does it always mean," but C doesn't say what it always means; it says what it can signify. That might be a trick. Let me see.

12. Perhaps C is the best answer because it refutes the idea that it always means one thing. Option C is not saying what it always means; it's saying that it can mean different things.

13. But the question asks for what it always means, implying a consistent meaning. In reality, there is no single thing that purring always means. It varies.

14. So, none of A, B, D are correct, but C is the only one that captures the variability. Option C starts with "A cat's purr can signify..." which is not claiming an always meaning, but it's the correct understanding.

15. Perhaps the question is designed to highlight that purring doesn't have a single always meaning, and C is the accurate choice.

16. Let me check online or recall reliable sources.

17. I think scientifically, purring is associated with both positive and negative situations.

18. For instance, cats purr when they're content but also when they're frightened or in pain.

19. So, it's not always hunger, not always stress, not always happy.

20. Therefore, C is the correct answer.

### Semantic Signals

1. Reasoning and Analysis

2. Reasoning and Analysis

3. Personal Opinion and Recall,

4. Examples and Illustration,

5. Personal Opinion and Recall,

6. Addition and Elaboration,

7. Conclusion and Summary,

8. Contrast and Concession,

9. Contrast and Concession,
10. Contrast and Concession,

11. Conclusion and Summary,

12. Personal Opinion and Recall,

13. Contrast and Concession,

14. Conclusion and Summary,

15. Personal Opinion and Recall,

16. Reasoning and Analysis,
17. Personal Opinion and Recall,
18. Examples and Illustration,

19. Conclusion and Summary,
20. Conclusion and Summary,

---

### Example-3

#### Question

Jan has three times the number of pets as Marcia. Marcia has two more pets than Cindy. If Cindy has four pets, how many total pets do the three have?

| Step-by-Step Reasoning Trace | Semantic Signals |
|---|---|
| 1. First, the question is: Jan has three times the number of pets as Marcia. Marcia has two more pets than Cindy. Cindy has four pets. I need to find the total number of pets for all three. Cindy has four pets. That's given directly: "Cindy has four pets." | 1. Reasoning and Analysis, |
| 2. Now, Marcia has two more pets than Cindy. So, if Cindy has 4, then Marcia has 4 + 2 = 6 pets. | 2. Reasoning and Analysis, |
| 3. Let me write that down: - Cindy: 4 pets - Marcia: Cindy's pets + 2 = 4 + 2 = 6 pets Next, Jan has three times the number of pets as Marcia. Marcia has 6 pets, so Jan has 3 times that. | 3. Reasoning and Analysis, |
| 4. So, Jan: 3 * Marcia's pets = 3 * 6 = 18 pets. | 4. Conclusion and Summary, |
| 5. Now, to find the total, I add up all their pets: Cindy: 4 Marcia: 6 Jan: 18 Total: 4 + 6 + 18 | 5. Reasoning and Analysis, |
| 6. Let me calculate that: 4 + 6 is 10, and 10 + 18 is 28. | 6. Reasoning and Analysis, |
| 7. So, the total should be 28 pets. I should double-check to make sure I didn't miss anything. The relationships: - Jan = 3 * Marcia - Marcia = Cindy + 2 - Cindy = 4. So, from Cindy, Marcia = 4 + 2 = 6 | 7. Conclusion and Summary, |
| 8. Then Jan = 3 * 6 = 18 Total = 4 + 6 + 18 = 28 I can set up equations to verify. Let C be Cindy's pets. C = 4, M = C + 2, J = 3 * M | 8. Conclusion and Summary, |
| 9. Then total T = J + M + C, Substitute: M = C + 2 = 4 + 2 = 6, J = 3 * M = 3 * 6 = 18, T = 18 + 6 + 4 = 28. Or directly: T = J + M + C = 3M + M + C = 4M + C | 9. Conclusion and Summary, |
| 10. But M = C + 2, so T = 4(C + 2) + C = 4C + 8 + C = 5C + 8, C = 4, so T = 5*4 + 8 = 20 + 8 = 28. Same thing. | 10. Contrast and Concession, |
| 11. So, it seems consistent. The question is to put the answer in a boxed format. | 11. Conclusion and Summary, |
| 12. Also, in my thinking, I should include keywords like "Actually," "Wait," etc. but I think I did it step by step without needing to backtrack, but I can add some for the reflection. | 12. Addition and Elaboration, |
| 13. For example, after finding Marcia has 6, I could say "OK, so Marcia has 6 pets." But it's fine. | 13. Examples and Illustration, |
| 14. I think the process is clear. | 14. Personal Opinion and Recall, |
| 15. So, the total number of pets is 28. | 15. Conclusion and Summary |

## H BROADER IMPACT

The development of Reasoning Scaffolding offers significant potential to reshape both the technical landscape of language modeling and the broader societal deployment of AI systems. By enabling small language models (SLMs) to internalize algorithmic reasoning structures rather than imitate surface-level text, our framework advances knowledge distillation toward greater logical robustness, interpretability, and resource efficiency. This approach directly addresses key challenges in AI safety and reliability, yielding SLMs that are less susceptible to spurious correlations and brittle failure modes, and facilitating trustworthy applications in domains such as education, healthcare, legal reasoning, and scientific discovery. By enhancing the reasoning abilities of compact models, Reasoning Scaffolding democratizes access to advanced AI, supporting robust performance on edge devices and in resource-limited environments. Furthermore, by shifting distillation from rote imitation to structured reasoning, our work paves the way for future research into more transparent and faithful knowledge transfer, including finer-grained reasoning signals, symbolic integration, and self-supervised techniques to further improve robustness and reduce reliance on proprietary LLMs.

