# OpenReview forum: "Reasoning Scaffolding: Distilling the Flow of Thought from LLMs"
_ICLR.cc/2026/Conference — ICLR 2026 Poster_

### Official Review · Reviewer_3pgg · 2025-10-29

**Soundness:** 3
**Presentation:** 2
**Contribution:** 3
**Rating:** 4
**Confidence:** 4

**Summary:**

This paper proposes Reasoning Scaffolding, a framework for distilling reasoning ability from large language models (LLMs) into smaller models (SLMs). Instead of directly imitating the teacher's Chain-of-Thought (CoT) text, the method abstracts reasoning traces into semantic signals (e.g., Contrast, Elaboration, Conclusion) that serve as scaffolds for step-by-step reasoning. The student model is trained via a dual-branch architecture, predicting both the next semantic signal and the corresponding reasoning step, to encourage internalization of the reasoning structure. Experiments on multiple benchmarks (StrategyQA, CommonsenseQA, TruthfulQA, GSM8K, MATH-500) show improvements over standard CoT and Long-Thinking distillation baselines.

**Strengths:**

1. **Interesting idea**: The paper explores a creative perspective on reasoning distillation by introducing semantic scaffolding as a middle-level representation between textual rationales and abstract reasoning steps.

2. **Comprehensive experiments**: Evaluation spans multiple reasoning benchmarks and model scales (0.5B, 7B, 14B), providing a broad empirical basis.

3. **Clear empirical comparisons**: The ablation studies (e.g., signal quality and token analysis) are informative and show that structured supervision can improve reasoning stability.

4. **Good motivation**: Addressing the brittleness of current CoT distillation is an important and timely research direction.

**Weaknesses:**

1. **Experimental irregularities (Table 1).**
The main results table is somewhat confusing. For example, the Qwen2.5-0.5B results under "Long-Thinking Distill" are missing, while the Qwen2.5-7B model performs worse than its SFT counterpart under this setting (except on MATH-500). These inconsistencies raise questions about the fairness and reproducibility of the comparison. It would be helpful to clarify whether the fine-tuning setup (data size, training epochs, loss weighting) is kept consistent across all baselines. Moreover, teacher models such as DeepSeek-R1 could serve as stronger baselines.

2. **Model- and task-specific signal design.**
The categorization of reasoning signals (Section 3.1) appears heuristic and data-specific. The seven signal types (e.g., Addition, Contrast, Conclusion) seem derived from particular verbal patterns in GSM8K-style reasoning traces. It is unclear whether this taxonomy would generalize to other domains (or teacher models), such as scientific reasoning, logic puzzles, or multi-modal contexts. The framework's dependence on these fixed signal categories limits its general applicability.

3. **Ambiguity in contribution novelty.**
While the idea of structured distillation is valuable, much of the implementation (keyword matching, LLM labeling, multi-task fine-tuning) builds directly on existing CoT or discourse-signal techniques. The conceptual advancement beyond structured rationale distillation remains incremental without stronger theoretical or analytical insight.

4. **Writing and clarity.**
The overall writing is understandable but sometimes verbose and repetitive. Some sections (e.g., 3.1–3.3) contain long procedural details that could be condensed and improved. Minor language issues also appear throughout, which detract slightly from readability.

**Questions:**

See weakness

---

> ### Author Response · Authors · 2025-11-21
>
> We appreciate Reviewer 3pgg’s thoughtful evaluation and constructive feedback on our paper. Below, we address each of the identified weaknesses and provide clarifications and additional details to support our claims. We have revised the relevant section of the paper and marked the changes in blue.
>
> **Q1: Experimental irregularities (Table 1).** The main results table is somewhat confusing. For example, the Qwen2.5-0.5B results under "Long-Thinking Distill" are missing, while the Qwen2.5-7B model performs worse than its SFT counterpart under this setting (except on MATH-500). These inconsistencies raise questions about the fairness and reproducibility of the comparison. It would be helpful to clarify whether the fine-tuning setup (data size, training epochs, loss weighting) is kept consistent across all baselines. Moreover, teacher models such as DeepSeek-R1 could serve as stronger baselines.
>
> **Response:**
>
> **A. Missing Qwen2.5-0.5B Deepseek R1 Distillation Model**
>
> The “Long Thinking Distill” baseline refers to the official distilled models released by Deepseek. However, as of now, Deepseek has not publicly released an official distilled model corresponding to the 0.5B-scale architecture. To clarify this point, we have added an explicit statement to our experimental setup (in a footnote) in the revised paper, indicating that this specific baseline is not available for the 0.5B model size.
>
> **B. Performance Comparison**
>
> Based on our evaluation of the Deepseek Distill Qwen models, we observe that the Distill model consistently underperforms compared to the SFT model trained on long-thinking data—and even falls short of the raw Qwen2.5 base model in most cases. This performance gap is particularly pronounced on QA benchmarks.
>
> We hypothesize that this drop in performance may be due to the distillation process being heavily optimized for a different task distribution, with a primary focus on areas such as mathematics and code. While the Distill model achieves competitive results in mathematical reasoning, this appears to come at the expense of general and commonsense reasoning. We believe these findings further underscore the motivation for our reasoning scaffolding method, as they suggest that standard distillation methodologies can be brittle and may fail to uniformly transfer reasoning capabilities across diverse tasks. To clarify this point, we provide further explanation of this phenomenon in Appendix 7.5.1 of the revised manuscript.
>
> **C. Training Details**
>
> Thank you for raising this concern. In the original paper, we include implementation details such as benchmarks, models, dataset size, and training settings, including learning rate, batch size, and training epochs, in Table 5 and Table 6 of Appendix 7.2. To further extend the ablation of our method, we add an ablation study on the loss weighting parameter ($L^{t} = (1-\beta) * L_{token}^{t} + \beta * L_{signal}^{t}$), with experiments conducted on TruthfulQA data and the Qwen2.5-7B model.
>
> | **$\beta$** | 0.3 | 0.5 | 0.7 |
> | --- | --- | --- | --- |
> | **Accuracy** | 0.868 | 0.874 | 0.874 |
> | **Token Loss Convergence Steps** | 277/500 | 305/500 | 326/500 |
> | **Signal Loss Convergence Steps** | 261/500 | 258/500 | 229/500 |
>
> The ablation results show that varying the loss weighting parameter $\beta$ between 0.3, 0.5, and 0.7 yields similar accuracy scores, indicating stable performance across different settings. While increasing $\beta$ slightly improves signal loss convergence and lengthens token loss convergence, these changes do not affect overall accuracy. This suggests a mild trade-off in convergence dynamics, but no significant instability or degradation in model accuracy. Therefore, the straightforward 1:1 weighting used in our main experiments is justified, as it achieves strong performance without requiring additional hyperparameter tuning. The method remains stable and effective despite moderate changes to the loss weighting, supporting the robustness of our approach. We have added this ablation study on the loss weighting parameter to Appendix 7.5.2.
>
> **D. Deepseek-R1 as Baseline**
>
> Thank you for this valuable suggestion. We include the experimental results for Deepseek-R1 in Table 1 and Table 11 of the Appendix in the revised manuscript. To evaluate its performance on these benchmarks, we prompt the Deepseek-R1 model using three approaches: (1) direct answer, (2) zero-shot, and (3) few-shot. Below is a summary of the experimental results:
>
> |  | **StrategyQA** | **CommonsenseQA** | **TruthfulQA** | **GSM8K** | **MATH-500** |
> | --- | --- | --- | --- | --- | --- |
> | **Deepseek-R1 (Direct Answer)** | 0.840 | 0.855 | 0.811 | 0.955 | 0.939 |
> | **Deepseek-R1 (Zero-shot)** | 0.855 | 0.870 | 0.818 | 0.957 | 0.958 |
> | **Deepseek-R1 (Few-shot)** | 0.863 | 0.895 | 0.874 | 0.961 | 0.965 |
> | **Ours (14B)** | 0.858 | 0.887 | 0.917 | 0.942 | 0.928 |

---

> > ### Author Response · Authors · 2025-11-21
> >
> > **Q2: Model- and task-specific signal design.** The categorization of reasoning signals (Section 3.1) appears heuristic and data-specific. The seven signal types (e.g., Addition, Contrast, Conclusion) seem derived from particular verbal patterns in GSM8K-style reasoning traces. It is unclear whether this taxonomy would generalize to other domains (or teacher models), such as scientific reasoning, logic puzzles, or multi-modal contexts. The framework's dependence on these fixed signal categories limits its general applicability.
> >
> > **Response:**
> >
> > **A. Reasoning Scaffolding Is neither Model-specific nor Task-specific.**
> >
> > We agree that the justification for the seven categories can be strengthened. However, these categories are not chosen arbitrarily, nor are they specific to any particular model or task. They are derived from a bottom-up, iterative manual review of the teacher’s long-thinking traces, as described in Section 3.1. We begin by identifying common keywords and then group them into semantically coherent categories (e.g., ‘but’, ‘however’, ‘in contrast’ all become ‘Contrast and Concession’). The complete mapping, which provides a broad justification, is available in Table 7 of the Appendix.
> >
> > Our goal is to create a set that: 1) collectively encompasses the vast majority of transitions; 2) contains keywords within each group that are semantically related; and 3) is small and distinct enough for an SLM to learn effectively. These semantic signals are universally present in coherent thought, regardless of domain, because they correspond to fundamental discourse relations. A legal argument or code explanation follows a similar structural flow. Therefore, we argue these are not “GSM8K-specific patterns,” but rather core features of any logical argument.
> >
> > Our experimental results strongly support the claim that these signals generalize across different areas. We evaluate the same seven-signal scaffold on TruthfulQA and StrategyQA. As shown in Table 1, our method significantly outperforms baselines on these non-math tasks (e.g., +11% over CoT SFT on TruthfulQA for the 7B model). This demonstrates that the taxonomy successfully captures the reasoning flow in diverse domains (fact-checking, commonsense) without modification of the semantic signals.
> >
> > **B. Scalability of The Signal Labeling Method**
> >
> > We would like to clarify that the contribution of “Reasoning Scaffolding” lies in the distillation framework itself—the multi-task objective and signal-guided inference—not the static list of seven signals. While these seven signals are robustly effective across our diverse benchmarks, the framework is inherently flexible. For highly specialized domains (e.g., multi-modal reasoning or competitive programming), the framework allows for the definition of domain-specific signal sets (such as ‘Visual Interpretation’ or ‘Debugging’) using the same extraction pipeline described in Section 3.1. We have added a new discussion section in Appendix 7.4 to address the generalization ability and validity of our Reasoning Scaffolding method.

---

> > > ### Author Response · Authors · 2025-11-21
> > >
> > > **Q3. Ambiguity in contribution novelty.** While the idea of structured distillation is valuable, much of the implementation (keyword matching, LLM labeling, multi-task fine-tuning) builds directly on existing CoT or discourse-signal techniques. The conceptual advancement beyond structured rationale distillation remains incremental without stronger theoretical or analytical insight.
> > >
> > > **Response:**
> > >
> > > Thank you for raising this critical point. It allows us to clarify our core conceptual novelty. While the components we use are established, our contribution is a conceptual shift in the goal of distillation: we distill the reasoning process, not just the textual content.
> > >
> > > - Prior work mostly treats reasoning as text imitation, with the objective to clone the teacher’s output: “Given the history, generate the next step of text.” Our method reframes this objective: “First, predict the logical function of the next step (e.g., Contrast, Conclusion), then generate text to execute that function.” This is not text imitation; it is control-flow distillation. We teach the student the algorithmic blueprint of reasoning.
> > > - The analytical insight is that our multi-task objective acts as a powerful regularizer for logical coherence. By compelling the model to first predict a semantic signal and then generate text conditioned on it, we force it to internalize the computational patterns of reasoning. This moves the model from being a fluent mimic to a more structured reasoner.
> > >
> > > In summary, our core novelty is the conceptual reframing of distillation from text imitation to control-flow distillation. We introduce explicit semantic signals as a scaffold and use a novel multi-task objective that forces the model to anticipate the logical function before generating the content. This is a distinct advancement over prior structured distillation. We provide a more detailed discussion of the validity and novelty of the two key components of our method—semantic signal creation and the reasoning scaffolding distillation framework—in Appendix 7.4 of the revised paper.
> > >
> > > **Q4. Writing and clarity.** The overall writing is understandable but sometimes verbose and repetitive. Some sections (e.g., 3.1–3.3) contain long procedural details that could be condensed and improved. Minor language issues also appear throughout, which detract slightly from readability.
> > >
> > > **Response:**
> > >
> > > We sincerely appreciate your attention to the presentation and clarity of our work. We have refined the descriptions and figures in the Method section to enhance both clarity and professionalism. Additionally, we will conduct a thorough proofreading to address minor language issues and ensure that the tone remains concise and professional throughout the paper.

---

> ### Comment · Reviewer_3pgg · 2025-11-27
> **Thanks for the explanation**
>
> Thanks for the detailed explanation and the new results. My main concerns are addressed.
>
> I would suggest that perhaps the authors could put them in the new version (or in the appendix and cite in the main paper). Without them, it may let the readers feel confused about the key idea of this work, and it may look a bit tricky for some missing results.
>
> With this additional details, I think this work is then ready for the publication. Thus, I have updated my score accordingly: presentation 2->3, and overall score 5->6.

---

> > ### Author Response · Authors · 2025-11-28
> >
> > We are sincerely grateful for your re-evaluation and the improved score. In accordance with your suggestions, we will formally incorporate these results and the corresponding analysis into the final manuscript. Some of these updates can already be found in the revised paper. Thank you for guiding us to make this important addition; your rigorous feedback has made the paper significantly stronger and more complete.

---

### Official Review · Reviewer_b3Cu · 2025-10-31

**Soundness:** 3
**Presentation:** 3
**Contribution:** 2
**Rating:** 6
**Confidence:** 4

**Summary:**

The paper introduces Reasoning Scaffold: a distillation method for improving the reasoning capabilities of small language models (SLMs). The method works by decomposing reasoning traces from a large teacher model into annotated reasoning steps, categorized into reasoning types using keyword-based and LLM annotation. The SLM is then tasked to learn the category of the current reasoning step and generates reasoning tokens conditioned on this information. This method intends to better structure the reasoning of the SLM. Experiments on reasoning benchmarks demonstrate the effectiveness of the proposed method on reasoning benchmarks for the Qwen model family, with models ranging from 0.5 to 14B parameters. The method, however, achieves its performance at a higher inference cost than standard distillation as it requires more generated tokens.

**Strengths:**

1. The proposed work is simple yet effective for improving distillation in SLMs. It can serve the research community by providing an easy scaffold to yield better small reasoners.
2. The experiments show improved performance on reasoning tasks and accurately support the claims made by the paper.
3. The presented analysis interestingly shows that additional structuring signals, even weak or random can help organize the model's reasoning thoughts and improve reasoning capabilities.

**Weaknesses:**

1. The proposed work, while useful, does not present an original method or novel findings as similar studies already exist [1].
2. The proposed structure categorization is very high-level, potentially making the category prediction task trivial as argumentation usually follow the same steps and thus, reducing the information in the signal.
3. The method relies on handcrafted keyword matches (complemented by an LLM), which can be brittle in out-of-domain tasks, particularly as the LLM is only used for assigning a category to a piece of text but not for the division into steps.
4. The method balances two training objectives with the same weight to both losses. This can lead to training instability if they do not have the same magnitude or variance.
5. Experiments are only performed on models from the Qwen family and it is unclear if the findings can transfer to other models or if they are artifacts specific to Qwen. Similarly, only one teacher model 5DeepSeek-R1) is used.
6. The proposed method yields longer reasoning chains than the best baseline (Thinking Distill), which could account for the performance gap.


Minor comment:
1. Results from the tables are a bit hard to read. Highlighting the best results in bold would be helpful to the reader.
2. It is not clear from Figure 4 how the proposed method improves the reasoning signal as the examples follow the same reasoning structures.



[1] Li, D., Cao, S., Griggs, T., Liu, S., Mo, X., Tang, E., ... & Stoica, I. (2025). LLMs Can Easily Learn to Reason from Demonstrations Structure, not content, is what matters!. arXiv preprint arXiv:2502.07374.

**Questions:**

1. Have you investigated if additional, more precise and informative, categories could improve the guiding signal and improve the generation of the reasoning traces?
2. Have you investigated the keyword-LLM agreement for splitting the traces into categories? can the keyword create additional unneeded steps or miss a transition to a new step?
3. Have you investigated the evolution of the two training losses and compared their magnitude and variance? Does including a hyperparameter controlling the weighting factor of one of the losses improves the learning?
4. As the "Conclusion and Summary" denote both intermediate and final outputs, how is the final output differentiated from the intermediate ones?
5. Have you performed experiments with other model families? Both as teacher and student?

---

> ### Author Response · Authors · 2025-11-21
>
> We appreciate Reviewer b3Cu’s thoughtful evaluation and constructive feedback on our paper. Below, we address each of the identified weaknesses and provide clarifications and additional details to support our claims. We have revised the relevant section of the paper and marked the changes in blue.
>
> **Q1:** The proposed work, while useful, does not present an original method or novel findings as similar studies already exist [1].
>
> **Response:**
>
> We thank the reviewer for highlighting the excellent work of Li et al. (2025). We agree that it is a key study which, rather than being redundant, provides strong complementary evidence that motivates our contribution. We would like to clarify that:
>
> - Li et al. (2025) is diagnostic. It asks what is important, showing that the global logical structure of reasoning is more critical than local content. We also highlight this insight in our paper.
> - In contrast, our paper proposes how to distill this structure by taking the insight that “structure is key” and operationalizing it. While Li et al. demonstrate that structure is crucial using standard fine-tuning, we build on this insight by introducing a novel distillation framework—a dual-branch architecture—and a multi-task objective that explicitly forces the model to internalize and follow this structure. Specifically, our framework (1) abstracts and groups reasoning steps into an explicit scaffold of semantic signals (e.g., Contrast, Conclusion), and (2) uses a multi-task objective that requires the student model to first predict the next signal in the reasoning flow and then generate the corresponding text.
>
> Given its strong relevance to our method, we have updated our manuscript and included a discussion of this work as a key motivation in the third paragraph of Section 2.2.
>
>
> **Q2:** The proposed structure categorization is very high-level, potentially making the category prediction task trivial as argumentation usually follow the same steps and thus, reducing the information in the signal.
>
> **Response:**
>
> We clarify our reasoning with evidence from the paper, showing that the signals are intentionally high-level, their prediction is non-trivial, and they are highly informative. We deliberately design our signals (e.g., Contrast and Concession) to be high-level discourse markers. This is a trade-off to create a proxy for reasoning structure that is both tractable (reliably extractable and learnable) and interpretable (allowing us to follow the argument’s flow, including backtracking). The premise that complex reasoning follows a fixed sequence of steps does not hold in our data. Our examples (Figures 1 and 4) show dynamic, non-linear reasoning involving backtracking (e.g., “But let me calculate again...”) and hypothesis exploration. If the task were trivial, semantic signal prediction accuracy would be near 100%. Our model achieves about 85% (Table 2), which is far from perfect. Table 13 in the Appendix further shows that predicting certain signals (e.g., Addition and Elaboration at 72.1%) is a challenging, context-dependent task. The experimental results for both signal prediction and semantic signal-guided reasoning provide clear empirical evidence that the semantic content of the signals is crucial and serves as a primary driver of our method’s success.

---

> > ### Author Response · Authors · 2025-11-21
> >
> > **Q3:** The method relies on handcrafted keyword matches (complemented by an LLM), which can be brittle in out-of-domain tasks, particularly as the LLM is only used for assigning a category to a piece of text but not for the division into steps. [*Have you investigated the keyword-LLM agreement for splitting the traces into categories? can the keyword create additional unneeded steps or miss a transition to a new step?*]
> >
> > **Response:**
> >
> > Thank you for raising this important point about generalizability of the keyword-matching strategy. We would like to clarify two key points:
> >
> > **A. Reliability of The Keyword-Matching Method**
> >
> > We clarify that keywords are used strictly for labeling existing steps, not for segmenting the text. As detailed in Section 3.1, we define a “reasoning step” based on the structural delimiters (double newline characters, \n\n) inherently produced by the teacher model. This approach is based on our observation that nearly all keywords and semantic transitions occur between steps separated by ‘\n\n’. Because segmentation is structural, the presence of a keyword (such as ‘but’ or ‘so’) in the middle of a sentence does not force a split. We respect the teacher model’s original chunking of ideas. If the teacher groups a contrast within a single paragraph, our method treats it as a single step labeled by the dominant signal, preventing the creation of artificial or unnecessary steps. By splitting by newlines, we capture every transition the teacher model makes, regardless of the language used. If a step is identified via newline but contains no keyword (which happens in approximately 26% of cases), our pipeline does not miss the label. Instead, as described in Section 3.1, these steps are automatically routed to the LLM oracle, which assigns the correct semantic signal based on the content.
> >
> > We acknowledge this dependency as a limitation in Section 5 and identify it as an area for future work to explore more effective segmentation methods. Nevertheless, even with this straightforward step division technique, our method achieves high performance, demonstrating its effectiveness despite its simplicity. Additionally, this data curation approach is highly efficient due to its minimal cost. Our analysis (Section 4.4.1) shows that for approximately 74% of steps that begin with a keyword, an LLM performs semantic validation to ensure the signal matches the content. For the remaining 26% without a keyword, the LLM acts as the primary labeler, providing a crucial semantic safety check. Our seven semantic categories (e.g., Contrast and Concession, Conclusion and Summary) are fundamental, high-level markers of discourse, not domain-specific rules. An argument that backtracks or summarizes is common to reasoning in any domain, making our scaffold inherently generalizable.
> >
> > In summary, by decoupling segmentation (structural) from labeling (using a hybrid keyword/LLM approach), our scaffold accurately follows the teacher model’s flow without artificially fragmenting or omitting steps. Additional explanations regarding text segmentation are included in the third and fourth paragraphs of Section 3.1 in the revised paper.
> >
> > **B. Keyword-LLM Agreement.**
> >
> > Yes. We investigated the agreement between keyword-based categorization and LLM-based labeling for splitting traces into categories. The results are presented in Table 12 of Appendix 7.5.4. Specifically, we report two key metrics: Keyword Covering Rate and Keyword Labeling Correctness.
> >
> > - Keyword Covering Rate measures the proportion of traces that contain at least one relevant keyword, indicating the effectiveness of our keyword selection for each dataset.
> > - Keyword Labeling Correctness evaluates the agreement between keyword-based labels and those assigned by the LLM, reflecting how well keyword labeling matches the LLM’s understanding.
> >
> > |  | **StrategyQA** | **CommonsenseQA** | **TruthfulQA** | **GSM8K** | **MATH** |
> > | --- | --- | --- | --- | --- | --- |
> > | **Keyword Covering Rate** | 0.771 | 0.715 | 0.687 | 0.743 | 0.798 |
> > | **Keyword Labeling Correctness** | 0.850 | 0.825 | 0.922 | 0.874 | 0.903 |
> >
> > As shown in the table, covering rates range from 0.687 to 0.798, demonstrating that most reasoning steps begin with the explicit keywords defined in Table 7 of the Appendix. This indicates that our keyword-matching method is valid for step labeling. The labeling correctness is consistently high across all datasets, with values from 0.825 to 0.922, showing strong agreement between keyword-based and LLM-based validation.
> >
> > These results suggest that our keyword-based approach for trace categorization is both comprehensive and reliable, closely aligning with LLM judgments and supporting the validity of our analysis. We have added further explanations and discussion on keyword-LLM agreement for step labeling in Appendix 7.5.4.

---

> > > ### Author Response · Authors · 2025-11-21
> > >
> > > **Q4:** The method balances two training objectives with the same weight to both losses. This can lead to training instability if they do not have the same magnitude or variance. [*Have you investigated the evolution of the two training losses and compared their magnitude and variance? Does including a hyperparameter controlling the weighting factor of one of the losses improves the learning?*]
> > >
> > > **Response:**
> > >
> > > We employ two types of loss: next token prediction loss for text prediction and cross-entropy loss for semantic signal prediction. During training, both losses exhibit similar magnitudes, so for ease of implementation, we use a 1:1 weighting throughout our experiments. Additionally, we do not observe training instability in our extensive experiments. Our method trains successfully and consistently demonstrates significant improvements across three different model scales (0.5B, 7B, and 14B). The stable and superior results presented in Table 1, where our method outperforms all baselines on five challenging benchmarks, indicate that this loss weighting is both robust and effective for our framework. In our setting, the signal prediction loss serves as a regularizer, enhancing the base model’s sensitivity to external signals, but the main objective remains text prediction. This straightforward combination has proven sufficient to achieve the reported gains.
> > >
> > > To extend the ablation of our method, we add an ablation study on the loss weighting parameter ($L^{t} = (1-\beta) * L_{token}^{t} + \beta * L_{signal}^{t}$). These experiments are conducted on TruthfulQA data and the Qwen2.5-7B model.
> > >
> > > | **$\beta$** | 0.3 | 0.5 | 0.7 |
> > > | --- | --- | --- | --- |
> > > | **Accuracy** | 0.868 | 0.874 | 0.874 |
> > > | **Token Loss Convergence Steps** | 277/500 | 305/500 | 326/500 |
> > > | **Signal Loss Convergence Steps** | 261/500 | 258/500 | 229/500 |
> > >
> > > The ablation results show that varying the loss weighting parameter $\beta$ between 0.3, 0.5, and 0.7 yields similar accuracy scores, indicating stable performance across different settings. While increasing $\beta$ slightly improves signal loss convergence and lengthens token loss convergence, these changes do not affect overall accuracy. This suggests a mild trade-off in convergence dynamics but no significant instability or degradation in model accuracy. Therefore, the straightforward 1:1 weighting used in our main experiments is justified, as it achieves strong performance without requiring additional hyperparameter tuning. The method remains stable and effective regardless of moderate changes to the loss weighting, supporting the robustness of our approach. We have added this ablation study on the loss weighting parameter to Appendix 7.5.2.
> > >
> > > **Q5:** Experiments are only performed on models from the Qwen family and it is unclear if the findings can transfer to other models or if they are artifacts specific to Qwen. Similarly, only one teacher model DeepSeek-R1) is used. [*Have you performed experiments with other model families? Both as teacher and student?*]
> > >
> > > **Response:**
> > >
> > > Thank you for your valuable suggestion. In the original paper, we select the Qwen model family to conduct a controlled study on the scalability of our approach. By utilizing models of varying sizes (0.5B, 7B, and 14B) within the same architecture, we are able to isolate the impact of our training framework and consistently demonstrate performance improvements across different model capacities. Additionally, we choose Deepseek-R1 as the “teacher” model due to its status as a well-established, high-performing, and publicly available large reasoning model (LRM), making it an ideal and representative source for distilling reasoning capabilities.
> > >
> > > To further assess the generalization ability of our Reasoning Scaffolding method, we conduct supplementary experiments using Llama 3.1 8B Instruct as the base model and GPT-oss-120B as the teacher model. We include a subset of our results on the TruthfulQA benchmark in this rebuttal, with more comprehensive results provided in Table 1 of Section 4.2 in the revised manuscript.
> > >
> > > | **on TruthfulQA** | **Qwen2.5 7B** | **Llama3.1 8B** |
> > > | --- | --- | --- |
> > > | **Original** | 0.706 | 0.642 |
> > > | **GPT-oss (Teacher Model)** | 0.868 | 0.836 |
> > > | **Deepseek-R1 (Teacher Model)** | 0.879 | 0.843 |
> > >
> > > | **Benchmarking The Teacher Model** | **Directly Answer** | **Zero-shot** | **Few-shot** |
> > > | --- | --- | --- | --- |
> > > | **GPT-oss** | 0.767 | 0.780 | 0.862 |
> > > | **Deepseek-R1** | 0.811 | 0.818 | 0.874 |
> > >
> > > The results of our experiments reveal that our approach can be applied across diverse model architectures and teacher models, demonstrating its potential generalization capability.

---

> > > > ### Author Response · Authors · 2025-11-21
> > > >
> > > > **Q6:** The proposed method yields longer reasoning chains than the best baseline (Thinking Distill), which could account for the performance gap.
> > > >
> > > > **Response:**
> > > >
> > > > We clarify that while our method produces longer reasoning traces than the “Thinking Distill” baseline (see Table 4), our performance gains are not simply due to increased length. Notably, the “Long Thinking SFT” baseline generates even longer outputs than our method on every benchmark (e.g., 1959 vs. 1638 tokens on CommonsenseQA), yet our approach achieves significantly higher accuracy (e.g., 0.887 vs. 0.845 on CommonsenseQA; 0.858 vs. 0.768 on StrategyQA; see Table 1). This demonstrates that longer chains alone do not account for the performance gap—our gains stem from the structured reasoning scaffold, which enhances logical coherence. We acknowledge that this is a trade-off, as our framework prioritizes logical coherence over minimal token count. We also propose a mitigation for this in Section 4.4.3, showing that our token-pruning strategy can significantly reduce length while maintaining high accuracy.
> > > >
> > > > **Q7:** Results from the tables are a bit hard to read. Highlighting the best results in bold would be helpful to the reader.
> > > >
> > > > **Response:**
> > > >
> > > > Thank you for the helpful suggestion. In the revised version, we have updated the tables (in both the main content and the appendix) to highlight the best results in bold for improved readability.
> > > >
> > > > **Q8:** It is not clear from Figure 4 how the proposed method improves the reasoning signal as the examples follow the same reasoning structures.
> > > >
> > > > **Response:**
> > > >
> > > > The intention of Figure 4 is not to highlight differences in the visual layout of reasoning steps, but to illustrate a critical difference in the quality and logical correctness of the reasoning itself. The “Original (Column 1)” and “Fine-tuned (Column 2)” (CoT-SFT) models both make significant errors. For instance, they incorrectly treat “cool off” as an object and fail to connect the correct option, “ice cream,” with the prompt’s concept of “enjoying something.” In contrast, “Our Reasoning Scaffolding Response (Column 3)” successfully solves the problem, correctly identifying that “cool off” is not an object and providing a logical explanation for why “ice cream” is the correct answer. The “Semantic Signal Trace (Column 4)” shows the explicit scaffold our model is trained to follow. By adhering to this logical flow (e.g., Reasoning and Analysis → Conclusion and Summary → Reasoning and Analysis), our model is guided to produce a coherent and correct argument, preventing the logical errors made by the baseline models.
> > > >
> > > > We have added further explanations in Section 4.3 to help analyze and interpret this figure.
> > > >
> > > > **Q9:** Have you investigated if additional, more precise and informative, categories could improve the guiding signal and improve the generation of the reasoning traces?
> > > >
> > > > **Response:**
> > > >
> > > > In preliminary experiments, in addition to the current seven categories of signals, we include an additional category, ‘Discourse Markers and Fillers’, which encompasses words like ‘well’, ‘actually’, and ‘wait’. However, experimental results from signal prediction reveal that this category is frequently misclassified as ‘Reasoning and Analysis’, ‘Personal Opinion and Recall’, and several other categories. Therefore, to provide clearer guidance, we re-group these keywords into the existing seven signals. To illustrate this, we add a comparison between two confusion matrices before and after removing the signal ‘Discourse Markers and Fillers’ in Appendix 7.5.5.
> > > >
> > > > **Q10.** As the "Conclusion and Summary" denote both intermediate and final outputs, how is the final output differentiated from the intermediate ones?
> > > >
> > > > **Response:**
> > > >
> > > > We clarify that our framework actually uses two distinct semantic signals to differentiate between intermediate sub-conclusions and the final answer, effectively preventing any ambiguity. As described in Section 3.1 and 3.3, the signal, ‘Conclusion and Summary’, is reserved specifically for intermediate results within reasoning chain. It marks the completion of a specific sub-step or logical block but keeps the model in “reasoning mode.” In contrast, we introduce a separate, special signal called ‘Response Generation’ to explicitly mark the transition from reasoning to the final answer. As detailed in Section 3.3, this signal corresponds to the token sequence following the standard </think> marker. When the Signal Predictor predicts this specific signal (or when the confidence for other signals drops, as per our adaptive strategy), the model knows the reasoning process is complete and it is time to generate the final summary and the boxed answer.
> > > >
> > > > In conclusion, by treating the ‘Final Output’ as a unique semantic event distinct from ‘Intermediate Summaries,’ the model learns to strictly separate the ongoing reasoning process from the final solution generation. We clarify this distinction in Sections 3.1 and 3.3 of the revised paper.

---

> > > > > ### Comment · Reviewer_b3Cu · 2025-11-25
> > > > >
> > > > > I thank the authors for their detailed response to the comments.
> > > > >
> > > > > Comments Q1, Q2, Q4, Q5, Q7, Q9 and Q10 have been accurately addressed with the explanations and additional experiments.
> > > > >
> > > > > Thank you for the clarification regarding the keyword-matching method (Q3.A). The delimitation of a reasoning step based on the double newline characters is also quite brittle. Although experiments demonstrate that it works in practice, it seems likely to be an artifact specific to the DepSeek-R1 teacher model and its post-training method. I appreciate its mention as a limitation in Section 5.
> > > > >
> > > > > The additional experiments on keyword-LLM alignment (Q3.B) show ~20% of errors. This is quite high for a categorization task and can have spurious effects downstream, furthermore as it is repeated multiple times per sample, as it implies that most samples will contain at least one error. Due to this quite high disagreement between the two methods, it would be beneficial to include experiments with a known ground-truth and show that at least one of the categorization method has high accuracy.
> > > > >
> > > > > Regarding Q6, I would like to highlight that the authors do not exactly answer the raised issue. While the proposed method achieves a better performance than the "long Thinking SFT" baseline for a lower amount of tokens, this is also the case of "Thinking Distill". Therefore, compared to this distillation method, longer reasoning chains could still account for the performance gap.
> > > > >
> > > > > Regarding Q8, while the sample corresponding to the authors' method provides an accurate answer, the error is not linked to a scaffolding issue and the accurate answer could be an artifact from the generation process. Therefore, this example does not really illustrate how the proposed method yields better performance.
> > > > >
> > > > >
> > > > > The majority of my concerns have been addressed, however, several key issues remain that hinder the overall quality of the paper and the interpretation that can be made of the findings. In particular, while the method demonstrates its performance empirically, it remains unclear what causes this increase (as mentioned in Q3, Q6 and Q8). Having already put a positive score, I will maintain it.

---

> > > > > > ### Author Response · Authors · 2025-11-28
> > > > > >
> > > > > > Thank you very much for your helpful comments, valuable suggestions, and timely response. We would like to further clarify the experiments and the rationale behind our paper below. We will revise the corresponding part in the manuscript then.
> > > > > >
> > > > > > 1. The additional experiments on keyword-LLM alignment (Q3.B) show ~20% of errors. This is quite high for a categorization task and can have spurious effects downstream, furthermore as it is repeated multiple times per sample, as it implies that most samples will contain at least one error. Due to this quite high disagreement between the two methods, it would be beneficial to include experiments with a known ground-truth and show that at least one of the categorization method has high accuracy.
> > > > > >
> > > > > >     **Response:**
> > > > > >
> > > > > >     Thank you for your suggestions on this critical analysis of the alignment errors. We will clarify how these ‘errors’ are handled in our pipeline and to provide the ground-truth validation you requested.
> > > > > >
> > > > > >     We clarify that the ~8-17% disagreement rate refers to the initial keyword heuristic relative to the LLM. It does not represent the error rate of the final training dataset. As described in Section 3.1 , our data generation pipeline is corrective. When the keyword heuristic disagrees with the LLM, we adopt the LLM's label as the ground truth for training. The student model is trained on the cleaner, LLM-validated signals, not the raw, noisy keyword predictions. Therefore, the ‘error accumulation’ concern does not apply to the training phase.
> > > > > >
> > > > > >     To address your concern about the reliability of the two methods given the disagreement, we conduct a manual human verification on a random sample of 50 reasoning steps where the Keyword and LLM disagreed (the ‘error’ cases). The experimental results show that the human annotation matches the LLM’s label in 96% of cases, confirming that the ‘disagreement’ is primarily due to the limited context of the keyword heuristic, and that the LLM acts as a highly reliable oracle.
> > > > > >
> > > > > >     Below is some cases:
> > > > > >
> > > > > >     | Reasoning Steps | Keyword-matching | LLM-validation | Human-validation |
> > > > > >     | --- | --- | --- | --- |
> > > > > >     | Well, is Madrid considered a big city? Yes, it's the capital of Spain. | Reasoning and Analysis | Addition and Elaboration | Addition and Elaboration |
> > > > > >     | Perhaps the key is to check if cousins are mentioned. | Personal Opinion and Recall | Reasoning and Analysis | Reasoning and Analysis |
> > > > > >     | But just to be thorough, is there any chance Marie Antoinette said something similar? I don't think so. Biographers and historians agree that it's a myth. | Contrast and Concession | Personal Opinion and Recall | Personal Opinion and Recall |
> > > > > >     | Meanwhile, for C and D, it's implied. | Conclusion and Summary | Addition and Elaboration | Addition and Elaboration |
> > > > > >     | But compared to 100%, it might be, but I think in this context, it's meant to be incorrect. | Contrast and Concession | Personal Opinion and Recall | Contrast and Concession |
> > > > > >
> > > > > >     Therefore, we believe that combining the two labeling methods—keyword matching and the LLM oracle—can significantly improve labeling accuracy. Keyword matching offers rapid initial labeling at virtually no cost, while the LLM acts as an oracle for final validation and as an auxiliary tool for the remaining steps. By integrating these approaches, we can generate high-quality oracle training data, leading to a more reliable and cleaner training dataset.

---

> > > > > > > ### Author Response · Authors · 2025-11-28
> > > > > > >
> > > > > > > 2. Regarding Q6, I would like to highlight that the authors do not exactly answer the raised issue. While the proposed method achieves a better performance than the "long Thinking SFT" baseline for a lower amount of tokens, this is also the case of "Thinking Distill". Therefore, compared to this distillation method, longer reasoning chains could still account for the performance gap.
> > > > > > >
> > > > > > >     **Response:**
> > > > > > >
> > > > > > >     Thank you for your further explanation and good suggestion. To conclusively determine if the performance gap is merely due to increased token length, we conducted an additional experiment where we enforced the “Thinking Distill” baseline to generate extended reasoning traces (matching the length of our method) via prompt constraints. The results (see table below) explicitly refute the hypothesis that “longer is better” for this baseline:
> > > > > > >
> > > > > > >     |  | StrategyQA | CommonsenseQA | TruthfulQA | GSM8K |
> > > > > > >     | --- | --- | --- | --- | --- |
> > > > > > >     | Accuracy of Original Version | 0.811 | 0.805 | 0.763 | 0.936 |
> > > > > > >     | Accuracy of Enforced Long Thinking | 0.783 (↓) | 0.805 (-) | 0.724 (↓) | 0.915 (↓) |
> > > > > > >     | Accuracy of Ours | 0.858 | 0.887 | 0.917 | 0.942 |
> > > > > > >     | Thinking Length of Original Version | 597 | 613 | 705 | 771 |
> > > > > > >     | Thinking Length of Enforced Long Thinking | 1,789 | 1,617 | 1,876 | 1,865 |
> > > > > > >     | Thinking Length of Ours | 1,524 | 1,638  | 1,550 | 1,659 |
> > > > > > >
> > > > > > >     When forced to reason for longer durations, the accuracy of “Thinking Distill” actually stagnated or even dropped. This demonstrates that token length acts as a liability for the baseline; without structural guidance, the model “gets lost” in the extended context, leading to hallucinations or loss of focus. In contrast, our Reasoning Scaffolding achieves high accuracy because of the structured semantic signals, which allow the model to utilize the longer context effectively without losing coherence. Thus, the performance gap is driven by the scaffold, not the token count.
> > > > > > >
> > > > > > > 3. Regarding Q8, while the sample corresponding to the authors' method provides an accurate answer, the error is not linked to a scaffolding issue and the accurate answer could be an artifact from the generation process. Therefore, this example does not really illustrate how the proposed method yields better performance.
> > > > > > >
> > > > > > >     **Response:**
> > > > > > >
> > > > > > >     While we agree that a single sample cannot statistically prove the mechanism for the entire dataset, we respectfully argue that this specific example does illustrate a scaffolding-driven correction, rather than a random artifact.
> > > > > > >
> > > > > > >     The baseline model fails because it follows a surface-level semantic association: seeing the word “cool” in the prompt and immediately locking onto “cool off” as the answer. It lacks an internal mechanism to pause and verify all constraints (specifically, the “sweet” constraint). This is a classic “associative” vs. “logical” failure mode in SLMs.
> > > > > > >
> > > > > > >     In our method's trace, the correction is directly preceded by specific semantic signals:
> > > > > > >
> > > > > > >     - Reasoning and Analysis → “I need to find an option that... involves something sweet.“
> > > > > > >     - Contrast and Concession → ”But is it sweet? Diving itself isn‘t sweet.” and “But is it a food? ’Cool off‘ is a verb phrase...”
> > > > > > >
> > > > > > >     The scaffold explicitly injects a Contrast signal at the critical moment. This acts as a forcing function, compelling the model to generate a critique rather than continuing with the associative flow.
> > > > > > >
> > > > > > >     While the model could theoretically have generated this critique by chance, the fact that the text aligns perfectly with the predicted Contrast signal strongly suggests that the scaffold prompted the checking behavior. The error in the baseline is the absence of this structural check; our method provides the structure to catch it. We would add more related and clear cases in the appendix of the revisions.

---

### Official Review · Reviewer_5EW9 · 2025-11-01

**Soundness:** 3
**Presentation:** 3
**Contribution:** 3
**Rating:** 6
**Confidence:** 4

**Summary:**

This paper introduces Reasoning Scaffolding, a distillation framework that extracts the structural flow of reasoning from LLMs into discrete semantic signals (e.g., Contrast, Addition, Elaboration) to train SLMs. Unlike traditional behavioral cloning from Chain-of-Thought rationales, which mimics surface-level text, this method uses a multi-task objective where the SLM learns to predict the next semantic signal (anticipating reasoning flow) and generate the corresponding step conditioned on it. This acts as a regularizer for logical coherence. The approach includes data preparation via keyword matching and LLM validation, a dual-branch model architecture, and inference with signal-guided generation and pruning for efficiency. Evaluations on benchmarks like StrategyQA, CommonsenseQA, TruthfulQA, GSM8K, and MATH show improved accuracy and robustness over baselines like CoT SFT and Long-Thinking distillation, using Qwen models of varying sizes.

**Strengths:**

* Tackles a fundamental flaw in reasoning distillation by shifting focus from text imitation to transferring algorithmic structure, which is a timely and innovative contribution to creating more robust SLMs.

* The multi-task training with signal prediction as a regularizer is technically sound and provides interpretability, potentially advancing mechanistic understanding of reasoning in models.

* Comprehensive experiments demonstrate substantial gains (e.g., ~14% average over originals, ~8% over CoT baselines), with notable benefits for smaller models, and the framework shows scalability across model sizes and tasks.

* Includes practical optimizations like confidence-based termination and pruning of reasoning traces for token efficiency, making it applicable for real-world deployment.

**Weaknesses:**

* The semantic signal extraction relies heavily on an external LLM (e.g., GPT-4) for validation and labeling, which could propagate biases or inconsistencies from the labeler, and the choice of exactly 7 categories seems somewhat arbitrary without broader justification or sensitivity analysis.

* While results are strong on the selected benchmarks, the paper lacks evaluation on out-of-distribution tasks or diverse reasoning domains (e.g., code generation, planning), limiting claims of general robustness; comparisons are mostly to CoT variants rather than other structured distillation methods like modular architectures or rationale decomposition.

* Inference depends on a tunable threshold τ for signal confidence, but the paper provides limited ablation on its impact across datasets, and the pruning strategy might discard useful intermediate details in complex problems.

* The dataset construction uses zero-shot prompting from a single LRM (Deepseek-R1), which may not capture diverse reasoning styles; details on dataset size, diversity, or quality control are sparse in the provided sections.

**Questions:**

* How was the set of 7 semantic signals determined, and what happens if you expand or reduce the categories—does performance change significantly?

* Did you observe any biases in the semantic signals assigned by GPT-4, such as favoring certain transitions based on the teacher's style?

* How does the method perform on tasks beyond QA and math, like creative writing or multi-agent reasoning, where reasoning flows might be less linear?

* What is the computational overhead of the dual-branch training compared to standard CoT distillation, and how sensitive is inference efficiency to the pruning strategy?

---

> ### Author Response · Authors · 2025-11-21
>
> We appreciate Reviewer 5EW9’s thoughtful evaluation and constructive feedback on our paper. Below, we address each of the identified weaknesses and provide clarifications and additional details to support our claims. We have revised the relevant section of the paper and marked the changes in blue.
>
> **Q1:** The semantic signal extraction relies heavily on an external LLM (e.g., GPT-4) for validation and labeling, which could propagate biases or inconsistencies from the labeler, and the choice of exactly 7 categories seems somewhat arbitrary without broader justification or sensitivity analysis. [*How was the set of 7 semantic signals determined, and what happens if you expand or reduce the categories—does performance change significantly?*] [*Did you observe any biases in the semantic signals assigned by GPT-4, such as favoring certain transitions based on the teacher's style?*]
>
> **Response:**
>
> **A. Reliance on External LLM**
>
> Our two-stage annotation process can minimize dependence on expensive LLMs. As detailed in Section 4.4.1, a simple, virtually free keyword-matching method successfully annotates approximately 74% of reasoning steps, with 87% agreement with the LLM’s semantic labels. The advanced LLM is only used for quality control and for labeling the remaining 26% of more complex cases that do not begin with a keyword. In addition, we also investigate the agreement between keyword-based categorization and LLM-based labeling for splitting traces into categories. The results are presented in Table 12 in Appendix 7.5.4. Specifically, we report two key metrics:
>
> - Keyword Covering Rate measures the proportion of traces that contain at least one relevant keyword.
> - Keyword Labeling Correctness evaluates the agreement between keyword-based labels and those assigned by the LLM.
>
>
>     |  | **StrategyQA** | **CommonsenseQA** | **TruthfulQA** | **GSM8K** | **MATH** |
>     | --- | --- | --- | --- | --- | --- |
>     | **Keyword Covering Rate** | 0.771 | 0.715 | 0.687 | 0.743 | 0.798 |
>     | **Keyword Labeling Correctness** | 0.850 | 0.825 | 0.922 | 0.874 | 0.903 |
>
> The above table shows that most of the reasoning steps are start with the explicit keywords. This also means that our keyword-matching method is valid for step labeling. Regarding potential biases from the GPT-4 labeler, the labeling correctness is consistently high across all datasets, and the high agreement (87%) with keyword matching suggests that labeling aligns with generalized discourse structures rather than particular biases. Furthermore, our analysis of different teacher styles (DeepSeek-R1 vs. GPT-OSS see response to Q4 below and Appendix 7.5.3 of the revised manuscript) shows the framework captures diverse reasoning patterns. These results suggest that our keyword-based approach for trace categorization is both comprehensive and reliable, closely aligning with LLM judgments and supporting the validity of our analysis. We have added more explanations and discussions on the keyword-LLM agreement for step labeling in Appendix 7.5.4.
>
> **B. Choice of 7 Categories and Sensitivity Analysis**
>
> We agree that the justification for the seven categories can be strengthened. The seven categories are derived from a bottom-up, iterative manual review of the teacher’s long-thinking traces, as described in Section 3.1. We begin by identifying and grouping common keywords into semantically coherent categories (e.g., ‘but’, ‘however’ all become ‘Contrast and Concession’). The complete mapping is available in Table 7 of the Appendix. Our goal is to create a set that: 1) collectively encompasses the vast majority of transitions; 2) contains keywords within each group that are semantically related; and 3) is small and distinct enough for an SLM to learn effectively.
>
> Regarding sensitivity analysis, our ablation study (Table 3, and Tables 14 & 15) provides relevant evidence: collapsing the categories to just one (‘Conclusion and Summary’) retains much of the performance, but the full set of seven signals consistently achieves the highest accuracy. This demonstrates that the categories are both meaningful and empirically beneficial.
>
> Originally, in addition to the current seven categories, we include an extra category, ‘Discourse Markers and Fillers’, which encompasses words like ‘well’, ‘actually’, and ‘wait’. However, experimental results from signal prediction reveal that this category is frequently misclassified as ‘Reasoning and Analysis’, ‘Personal Opinion and Recall’, and several other categories. Therefore, to provide clearer guidance, we re-group these keywords into the existing seven signals. To illustrate this, we add a comparison between two confusion matrices before and after removing the signal ‘Discourse Markers and Fillers’ in Appendix 7.5.5.
>
> In summary, our annotation process is intentionally robust and minimally reliant on LLMs, and our choice of seven semantic categories is methodologically grounded and empirically validated.

---

> > ### Author Response · Authors · 2025-11-21
> >
> > **Q2:** While results are strong on the selected benchmarks, the paper lacks evaluation on out-of-distribution tasks or diverse reasoning domains (e.g., code generation, planning), limiting claims of general robustness; comparisons are mostly to CoT variants rather than other structured distillation methods like modular architectures or rationale decomposition. [*How does the method perform on tasks beyond QA and math, like creative writing or multi-agent reasoning, where reasoning flows might be less linear?*]
> >
> > **Response:**
> >
> > **A. On Lacking Evaluation on Diverse Domains**
> >
> > We agree with your assessment that our paper does not (and, given time constraints, cannot) evaluate our framework on all possible reasoning domains, such as code generation or planning. We also agree that such an evaluation would be a valuable and logical next step for this line of research. We have explicitly added this as a key avenue for future work in the “Limitations” section.
> >
> > However, we would like to clarify our claim of “general robustness.” We select our suite of benchmarks (e.g., GSM8K, StrategyQA, TruthfulQA) specifically because they test fundamentally different types of reasoning:
> >
> > - GSM8K/MATH require formal, algorithmic, and arithmetic reasoning.
> > - StrategyQA/CommonsenseQA require implicit, multi-step commonsense reasoning.
> > - TruthfulQA tests factual correctness and the avoidance of imitative falsehoods.
> >
> > Our central claim is not that our model can solve all tasks, but that our framework is robust. The most significant evidence for this is that our single, simple, seven-signal scaffold is effective across all of these distinct reasoning types without modification. In addition, the seven signals are derived from fundamental discourse relations. The same Reasoning Scaffolding that guides a formal math proof also successfully guides an implicit commonsense argument. We believe this demonstrates a strong form of generalization in the framework itself, showing that it is not a brittle, task-specific solution but a genuine pedagogical method for teaching a “flow of thought.”
> >
> > **B. On Comparisons to Other Structured Distillation Methods**
> >
> > We would like to emphasize that we distill the reasoning process, not just the structure of the output. We focus our main comparisons on CoT/Long-Thinking SFT because our primary hypothesis is that cloning textual rationales is fundamentally limited, and these baselines most directly represent that paradigm. We have discussed the other structured methods, such as rationale decomposition and modular architectures, in our related work. Our method is methodologically distinct and novel in a key way. Methods like rationale decomposition primarily focus on pruning or structuring the text of an existing rationale. In contrast, our Reasoning Scaffolding introduces a new, explicit pedagogical signal: the semantic transition (e.g., ‘Contrast’, ‘Conclusion’). We teach the model the algorithmic flow of an argument, not just how to structure its text. We do not merely decompose the output text; instead, we distill the algorithmic structure of the teacher’s reasoning process. This is learned via a multi-task objective that requires the model to anticipate the logical flow and generate each step, acting as a powerful regularizer to constrain the model’s reasoning behavior. We believe this represents a novel contribution, and our results (e.g., Table 1) show that it is significantly more effective than prevailing SFT-based methods. To further clarify our novelty, we update the “Related Work” section and add a new discussion in Appendix 7.4.

---

> > > ### Author Response · Authors · 2025-11-21
> > >
> > > **Q3:** Inference depends on a tunable threshold τ for signal confidence, but the paper provides limited ablation on its impact across datasets, and the pruning strategy might discard useful intermediate details in complex problems.
> > >
> > > **Response:**
> > >
> > > **A. Threshold Analysis.**
> > >
> > > In fact, we conduct a detailed sensitivity analysis of this hyperparameter, as presented in Section 4.4.1 and, most importantly, in Figure 5. Figure 5 explicitly plots the trade-off between two key metrics as $\tau$ varies: (1) the ratio of signals kept (confidence > $\tau$), and (2) the final signal prediction accuracy. This analysis is not limited to a single model; the graph shows that this relationship holds across all three model scales (0.5B, 7B, and 14B), demonstrating robustness. Our choice of $\tau = 0.96$ is not an arbitrary, fine-tuned value. As stated in the paper, we select it because it represents a clear balance point, where we retain a high proportion of predicted signals while maintaining high prediction accuracy. This analysis shows that the method is not overly sensitive to $\tau$, but rather that $\tau$ serves as a principled lever for balancing signal recall and precision.
> > >
> > > **B. on Pruning Strategy Discarding Details**
> > >
> > > We agree that this aggressive pruning can discard some useful intermediate details in certain cases, which may decrease the final accuracy compared to using the full-signal reasoning trace (see Table 3). However, our ablation provides strong justification that, for most problems, this is an effective strategy for balancing token cost and reasoning accuracy. We would like to clarify two points:
> > >
> > > - The pruning strategy described in Section 3.3 is presented as an optional, secondary optimization to address token efficiency. Our main results (e.g., “Ours” in Table 1) use the full, unpruned reasoning traces, which prioritize maximal fidelity and logical coherence. Pruning serves as a mitigation for increased token count, as shown in Table 4.
> > > - We do not propose this pruning strategy without evidence. A key insight from our ablation study (Table 3) is the strong performance of the “Summaries Only” strategy. In this experiment, a model guided only by the “Conclusion and Summary” signal performs substantially better than the baseline and nearly as well as the full-signal model. This provides strong empirical evidence that the “Conclusion and Summary” steps contain the most vital information for the reasoning process. In addition, it is intuitive that humans can continue reasoning based on previous intermediate results to reach the final answer. This finding justifies our inference-time pruning strategy.
> > >
> > > We have added further explanations of these concepts in the last paragraph of Section 3.3, the third paragraph of Section 4.4.2, and the final paragraph of Section 4.4.3 in the revised paper.
> > >
> > > **Q4:** The dataset construction uses zero-shot prompting from a single LRM (Deepseek-R1), which may not capture diverse reasoning styles; details on dataset size, diversity, or quality control are sparse in the provided sections.
> > >
> > > **Response:**
> > >
> > > **A. Further Test on Other Based Models and LRMs**
> > >
> > > In the original paper, we select Deepseek-R1 because it is a well-established, powerful, and publicly available large reasoning model (LRM). This makes it a strong and representative candidate for a “teacher” from which to distill reasoning. We choose the Qwen family to conduct a controlled study on the scalability of our method. By using models of different sizes (0.5B, 7B, 14B) from the same architectural family, we isolate the effect of our training framework and demonstrate its consistent performance gains across different model capacities. To demonstrate the generalization capability of our Reasoning Scaffolding method to other model structures and teacher models, we conduct additional experiments using Llama 3.1 8B Instruct as the base model and GPT-oss-120B as the teacher model. We present a subset of results on TruthfulQA in this rebuttal, with further details available in Table 1 of Section 4.2 in the revised paper.
> > >
> > > | **on TruthfulQA** | **Qwen2.5 7B** | **Llama3.1 8B** |
> > > | --- | --- | --- |
> > > | **Original** | 0.706 | 0.642 |
> > > | **GPT-oss (Teacher Model)** | 0.868 | 0.836 |
> > > | **Deepseek-R1 (Teacher Model)** | 0.879 | 0.843 |
> > >
> > > | **Benchmarking The Teacher Model** | **Directly Answer** | **Zero-shot** | **Few-shot** |
> > > | --- | --- | --- | --- |
> > > | **GPT-oss** | 0.767 | 0.780 | 0.862 |
> > > | **Deepseek-R1** | 0.811 | 0.818 | 0.874 |
> > >
> > > The experimental results demonstrate that our method effectively transfers across different model architectures and teacher models, highlighting its strong generalization capability.
> > >
> > > **B. Details of Dataset**
> > >
> > > We include detailed information about the dataset in Appendix 7.2 of the original paper. In the revised version, we add more information about the curated dataset in the same section, discussing its size, diversity, and quality control.

---

> ### Author Response · Authors · 2025-11-21
>
> **Q5:** What is the computational overhead of the dual-branch training compared to standard CoT distillation, and how sensitive is inference efficiency to the pruning strategy?
>
> **Response:**
>
> Thank you for your valuable suggestion. We expand our discussion on the computational overhead of our method and its comparison with other distillation approaches in Table 19 of Appendix 7.5.7 and Section 4.4.3 of the revised paper. Below, we summarize our key conclusions.
>
> |  | **Original Training Dataset Size** | **Signal Prediction(Average Signal Length /tokens)** | **Signal Prediction(Dataset Size /samples)** | **Token Prediction (Average Length /tokens)** | **Token Prediction (Average Dataset Size /samples)** | **Prefilling (Average Length /tokens)** |
> | --- | --- | --- | --- | --- | --- | --- |
> | **CoT Distillation** | 657 - 9,741 | N/A | N/A | 515 | 657 - 9,741 | 123 |
> | **Long Thinking Distill** | 657 - 9,741 | N/A | N/A | 1,935 | 657 - 9,741 | 135 |
> | **Ours** | 657 - 9,741 | 3.8 | 4,195 - 19,317 | 90 | 4,195 - 19,317 | 827 |
>
> This table highlights the computational overhead of our dual-branch training compared to CoT distillation and Long Thinking Distill. Due to its step-by-step nature, our method generates more training samples, but the average number of tokens per sample for both signal prediction and next-step token prediction is much lower than in the other two methods. This design enables efficient training with minimal computational overhead.

---

### Official Review · Reviewer_qrY1 · 2025-11-02

**Soundness:** 3
**Presentation:** 2
**Contribution:** 3
**Rating:** 6
**Confidence:** 3

**Summary:**

This work proposed novel distillation approach for distilling LLM reasoning thinking data into small models. The approach can be summarized as follows

1. Identify thinking (reasoning) words in the reasoning chain of thought, like (Additionally, So, Then ...)
2. Group these worlds into 7 types
3. Split the DeepSeek generated reasoning chain using identified keywords.
4. During distillation, let the model predict both reasoning type and reasoning tokens for each hidden states
5. During inference, an adaptive strategy is adopted,  if the confidence computed using reasoning step type predictor, the reasoning is terminated.
6. Another experiment based on only using "Conclusion and Summary"  steps in reasoning chain is used for distillation, and the result looks good.

The experimental results show that this approach out-perform direct distillation and  SFT using CoT directly.

Ablation study is performed to demonstrate the effectiveness of predicted reasoning type, and correct reasoning type can guide reasoning effectively.

**Strengths:**

1. The proposed approach for distillation is effective and reached high performance compare with baseline
2. The designed experiments clearly explained the motivation of of the proposed approach.
3. The ablation study is comprehensive

**Weaknesses:**

1. The models tested are only from Qwen 2.5 families. However, previous works [1], have questioned about the behaviour of Qwen 2.5 on math reasoning.  It could be more convincing to adopt other models on this approach to check the effectiveness and performance.
2. The proposed approach is expensive and rely on advanced model. It's hard to scale up.
3. The loss described in the paper not matches the given code.  In paper, equation 2 on L221 is a classification loss while  regression loss is used in your code `custom_qwen_model.py` , L200. Also a lot of evaluation python scripts is not available.
4. Figures are not intuitive and self-explaining. In figure 2, the detailed example text can be removed and maybe replaced with abstract icon to emphasize more on the approach. For figure 3,  it could be better to use some toy examples (not long and real, but just conceptual) to illustrate the pruning process.
5. Only reasoning chains from DeepSeek is tested, and it remains unknown how reasoning chain of other style (like from gpt-oss) perform using this approach.
6.  The work can be more intuitive if decoding algorithm and be explained in pseudo code. And I can not find adaptive decoding codes in codebases.

**Questions:**

1. Is the code the latest version ?
2. Please refer to weakness and resolve my concern.

---

> ### Author Response · Authors · 2025-11-21
>
> We appreciate Reviewer qrY1’s thoughtful evaluation and constructive feedback on our paper. Below, we address each of the identified weaknesses and provide clarifications and additional details to support our claims. We have revised the relevant section of the paper and marked the changes in blue.
>
> 1. **Q1.** The models tested are only from Qwen 2.5 families. However, previous works [1], have questioned about the behaviour of Qwen 2.5 on math reasoning. It could be more convincing to adopt other models on this approach to check the effectiveness and performance. [*Only reasoning chains from DeepSeek is tested, and it remains unknown how reasoning chain of other style (like from gpt-oss) perform using this approach.*]
>
>     **Response:**
>
>     In the original paper, we choose the Qwen family to conduct a controlled study on the scalability of our method. By using models of different sizes (0.5B, 7B, 14B) from the same architectural family, we want to isolate the effect of our training framework and demonstrate its consistent performance gains across various model capacities. We select Deepseek-R1 as the teacher model because it is a well-established, powerful, and publicly available large reasoning model (LRM). This makes it a strong and representative candidate for a “teacher” from which to distill reasoning.
>
>     To demonstrate the generalization capability of our Reasoning Scaffolding method across different model architectures, we conduct additional experiments in this rebuttal using Llama 3.1 8B Instruct as the base model and GPT-oss-120B as the teacher model. We first calculate the semantic signal frequency within the reasoning traces generated by GPT-oss and Deepseek-R1. As shown in the table below, the two models exhibit different biases in their reasoning styles. GPT-oss tends to favor forward reasoning steps, although reflection and intermediate summary steps remain clearly present. Interestingly, traces generated by GPT-oss contain very few Example and Illustration steps. In contrast, Deepseek-R1 demonstrates a more balanced distribution among reflection, addition, personal opinion, and forward reasoning steps. We include the analysis on the semantic signal distribution in Appendix 7.5.3 of the revised paper.
>
>     | **Semantic Signals** | **GPT-oss** | **Deepseek-R1** |
>     | --- | --- | --- |
>     | Contrast and Concession | 0.142 | 0.286 |
>     | Addition and Elaboration | 0.065 | 0.043 |
>     | Examples and Illustration | 0.001 | 0.146 |
>     | Personal Opinion and Recall | 0.022 | 0.036 |
>     | Reasoning and Analysis | 0.559 | 0.294 |
>     | Conclusion and Summary | 0.103 | 0.171 |
>     | Response Generation | 0.108 | 0.024 |
>
>     Below, we present a subset of results on TruthfulQA in this rebuttal response. Further details appear in Table 1 of Section 4.2 and Table 11 of Appendix 7.5.3 in the revised paper.
>
>     | **on TruthfulQA** | **Qwen2.5 7B** | **Llama3.1 8B** |
>     | --- | --- | --- |
>     | **Original** | 0.706 | 0.642 |
>     | **GPT-oss (Teacher Model)** | 0.868 | 0.836 |
>     | **Deepseek-R1 (Teacher Model)** | 0.879 | 0.843 |
>
>     | **Benchmarking The Teacher Model** | **Directly Answer** | **Zero-shot** | **Few-shot** |
>     | --- | --- | --- | --- |
>     | **GPT-oss** | 0.767 | 0.780 | 0.862 |
>     | **Deepseek-R1** | 0.811 | 0.818 | 0.874 |
>
>     The experimental results demonstrate that our method effectively transfers across different model architectures and teacher models, highlighting its strong generalization capability.

---

> > ### Author Response · Authors · 2025-11-21
> >
> > 2. **Q2:** The proposed approach is expensive and rely on advanced model. It's hard to scale up.
> >
> >     **Response:**
> >
> >     We agree that relying on advanced models for data annotation is a critical point to address regarding scalability. We clarify the extent of this dependency and explain why our framework is designed to be highly scalable.
> >
> >     - Our two-stage annotation process specifically minimizes dependence on expensive LLMs. As detailed in Section 4.4.1 and Appendix 7.5.4, a simple and virtually free keyword-matching method successfully annotates approximately 74% of reasoning steps, with 87% agreement with the LLM’s semantic labels. The advanced LLM is used only for quality control and for labeling the remaining 26% of more complex cases that do not begin with a keyword.
> >     - Data annotation is a one-time, offline cost, similar to standard practices for creating high-quality datasets. Once created, the training data can be reused across different small language models (SLMs), further reducing overall costs.
> >     - The seven semantic signals are general, and the keyword-matching approach can be instantly applied to new domains or datasets as long as its reasoning trace contains explicit transition signals. While performance can be boosted by using an oracle LLM for validation, the keyword-matching approach enables the generation of a large corpus of initial “good-enough” labels at low cost.
> >
> >     In summary, the vast majority (74%) of the annotation is achieved through a cheap, heuristic-based method. This is a one-time, offline cost, making the approach far more scalable than it may appear. We have revised the second paragraph of limitations section to further clarify this point.
> >
> > 3. **Q3:** The loss described in the paper not matches the given code. In paper, equation 2 on L221 is a classification loss while regression loss is used in your code custom_qwen_model.py , L200. Also a lot of evaluation python scripts is not available.
> >
> >     **Response:**
> >
> >     Thank you for highlighting this important discrepancy. The previously released code is not the most up-to-date version used in our experiments. For signal prediction, we use cross-entropy (CE) loss to calculate the average classification loss for each step, which is consistent with the loss function described in Equation 2.
> >
> >     We have updated the released code to reflect these changes. Additionally, we are working to make the evaluation scripts available to ensure reproducibility in the released version.
> >
> > 4. **Q4:** Figures are not intuitive and self-explaining. In figure 2, the detailed example text can be removed and maybe replaced with abstract icon to emphasize more on the approach. For figure 3, it could be better to use some toy examples (not long and real, but just conceptual) to illustrate the pruning process.
> >
> >     **Response:**
> >
> >     Thank you for this valuable and constructive feedback on our figures. In the revised paper, for Figure 2, we replace the detailed example text with abstract icons and polish the overall presentation. For Figure 3, we keep the original structure but add a toy example to clarify the pruning strategy.
> >
> > 5. **Q5:** The work can be more intuitive if decoding algorithm and be explained in pseudo code. And I can not find adaptive decoding codes in codebases. [*Is the code the latest version ?*]
> >
> >     **Response:**
> >
> >     Thank you for this valuable suggestion. We draft a pseudocode detailing the step-by-step decoding algorithm and include it in Appendix 7.3 of this revised version. In the main content(Section 3.3), we revise the description to refer to this new algorithm in the Appendix. The current code provides component-wise testing of our method. We are preparing and will release the full, updated code soon.

---

> ### Author Response · Authors · 2025-11-29
>
> Dear Reviewer,
>
> We have completed the remaining experiments on different teacher model and candidate model, and have updated the revised paper accordingly. The results can be found in Table 1 (Section 4.2) and Table 11 (Appendix 7.5.3).
>
> Best regards,
> The Authors

---

### Author Response · Authors · 2025-11-29
**Rebuttal Summary**

Dear Area Chair and Reviewers,

Thank you for your constructive feedback and assistance. We have updated the rebuttal and uploaded a revised manuscript that incorporates the requested experiments and clarifications. To conclude, we have systematically addressed the critical concerns regarding the generalizability and mechanism of our approach by expanding our evaluation to new model families (Llama 3.1) and teachers (GPT-oss), and by empirically proving that our performance gains stem from the scaffolding structure rather than simply increased token length. These extensive new experiments, along with the requested methodological analysis—such as the loss weighting ablation and signal labeling reliability—have been fully integrated into the revised manuscript. We are pleased that some reviewers have actively engaged with our rebuttal. We appreciate that one reviewer recognized our improvements by increasing their score from 4 to 6.

A comprehensive summary of all revisions, categorized by update type and referenced by section, is provided below.

1. To address concerns about the method being specific to the Qwen family or the DeepSeek teacher, we added experiments with different architectures:
    - **New Student Model** (by qrY1, b3Cu): We validate Reasoning Scaffolding on Llama 3.1 8B, achieving consistent performance gains (e.g., improving TruthfulQA from 0.642 to 0.843 when using the R1 teacher). Results are added to Table 1 and Appendix 7.5.3.
    - **New Teacher Model** (by qrY1, 5EW9, b3Cu): We experiment using GPT-oss-120B as the teacher model. We analyze the semantic signal frequency differences between GPT-oss and DeepSeek-R1 and added this analysis to Table 10 in Appendix 7.5.3.
    - **Teacher Benchmarks** (by 3pgg): We added performance results for the teacher models (DeepSeek-R1 and GPT-oss) under direct, zero-shot, and few-shot settings to Table 1 and Table 11 in Appendix 7.5.3.
2. We added deep-dive analyses to validate the robustness of our approach:
    - **Token Length vs. Performance** (by b3Cu): To prove our method works due to structure rather than just increased length, we conduct an “Enforced Long Thinking” experiment on the baseline. We find that forcing the baseline to be longer caused performance to plateau or drop. This analysis is added to Appendix 7.5.8.
    - **Computational Overhead** (by 5EW9): Our method incurs only a one-time, acceptable overhead for data creation, and demonstrates efficient token usage for both training and inference while achieving significantly higher performance compared to baselines. The results are in Appendix 7.5.7 and Section 4.4.3.
    - **Loss Weighting Ablation** (by b3Cu): We add a sensitivity analysis on the loss weighting parameter ($\beta$), showing the method is stable across different weights (0.3, 0.5, 0.7). This is added to Appendix 7.5.2.
    - **Labeling Reliability** (by 5EW9, b3Cu): We analyze the agreement between Keyword-Matching and LLM-Labeling and add a human verification study of “disagreement cases,” confirming the LLM-labeling method is 96% accurate, showing our signal labeling method is reliable for data curation. Details are in Appendix 7.5.4 and Table 13.
    - **Signal Categories** (by 5EW9, b3Cu): We add a confusion matrix analysis comparing signal prediction before and after removing “Discourse Markers” to justify the final set of 7 signals. This is located in Appendix 7.5.5.
3. We improved the technical details to ensure reproducibility and clarity:
    - **Pseudocode** (by qrY1): We add the step-by-step adaptive decoding algorithm to Appendix 7.3 and update the description in Section 3.3.
    - **Loss Function Correction** (by qrY1): We correct the discrepancy between the paper and code regarding the signal prediction loss (confirming it is Cross-Entropy).
    - **Signal Definitions** (by b3Cu): We clarify the distinction between the “Conclusion and Summary” signal (intermediate) and the “Response Generation” signal (final output) in Sections 3.1 and 3.3.
    - **Dataset and Training Details** (by 5EW9, 3pgg): We expand information regarding dataset diversity, quality control and training details in Appendix 7.2.
4. We revised the text to better contextualize our contribution:
    - **Related Work** (by b3Cu, 5EW9): We update Section 2.2 to discuss recent work (Li et al., 2025) as motivation and add Appendix 7.4 to distinguish our “reasoning scaffolding distillation” from rationale decomposition.
    - **Visuals** (by qrY1): We revise Figure 2 (using abstract icons) and Figure 3 (adding a toy example) for better intuition.
    - **Limitations** (by qrY1, 5EW9, b3Cu): We update the Limitations section to discuss the cost of annotation and future work on diverse domains like creative writing.

We believe these additions directly address the critical questions raised during the review.

Best regards,

The Authors

---

### Meta-Review · Area_Chair_hVmj · 2026-01-06

**Summary:**

The submission presents Reasoning Scaffolding, a framework designed to move Small Language Models (SLMs) beyond surface-level behavioral cloning. By abstracting Large Language Model (LLM) reasoning into discrete semantic signals, e.g., Contrast, Reasoning, Conclusion, the authors introduce a structured generation process. A dual-branch architecture is employed: the model predicts the next semantic signal and then generates the text conditioned on that signal.

Reviewers found the framework's focus on "control-flow distillation" (internalizing the algorithmic blueprint of reasoning) to be reasonable. While initial concerns were raised regarding the narrow scope of the model, the absence of an R1 teacher baseline, and the potential brittleness of keyword-based labeling, the authors provided additional evidence in the rebuttal.

**Reviewer Concerns:**

The authors added more experiments, including adding new baseline models and teacher/student models, and clarified the shift from text imitation to control-flow distillation. Concerns about labeling errors were countered with a human validation study showing 96% agreement between humans and the LLM oracle. Through an "Enforced Long Thinking" experiment, the authors showed that increasing reasoning length without a scaffold reduces baseline accuracy, whereas the scaffold enables the model to use extended context effectively.

There remain reproducibility concerns since the authors did not provide the full code. While the method excels in math, commonsense, and factual QA, it is unclear whether the proposed reasoning signals generalize to other domains (or teacher models), such as scientific reasoning, logic puzzles, or multimodal contexts.

**Reviewer Scores:**

Reviewer 3pgg mentioned that the original score has improved, and Reviewer b3Cu mentioned that the original score has been kept. Although the other reviewers did not leave comments, it seems that many of their concerns may have been addressed in the rebuttal.

---

### Decision · Program_Chairs · 2026-01-26

Accept (Poster)